# Duplicated antibiotic resistance genes reveal ongoing selection and horizontal gene transfer in bacteria

Rohan Maddamsetti [1,2], Yi Yao [1,2], Teng Wang[1,2], Junheng Gao[3], Vincent T. Huang [1,2], Grayson S. Hamrick [1,2,4], Hye-In Son[1,2] & Lingchong You [1,2,4,5] ✉

Horizontal gene transfer (HGT) and gene duplication are often considered as separate mechanisms driving the evolution of new functions. However, the mobile genetic elements (MGEs) implicated in HGT can copy themselves, so positive selection on MGEs could drive gene duplications. Here, we use a combination of modeling and experimental evolution to examine this hypothesis and use long-read genome sequences of tens of thousands of bacterial isolates to examine its generality in nature. Modeling and experiments show that antibiotic selection can drive the evolution of duplicated antibiotic resistance genes (ARGs) through MGE transposition. A key implication is that duplicated ARGs should be enriched in environments associated with antibiotic use. To test this, we examined the distribution of duplicated ARGs in 18,938 complete bacterial genomes with ecological metadata. Duplicated ARGs are highly enriched in bacteria isolated from humans and livestock. Duplicated ARGs are further enriched in an independent set of 321 antibiotic-resistant clinical isolates. Our findings indicate that duplicated genes often encode functions undergoing positive selection and horizontal gene transfer in microbial communities.

Selection for higher gene expression can promote the rapid evolution of duplicated genes through diverse molecular mechanisms[1–5]. Furthermore, gene duplication has long been recognized as a crucial step in the evolution of new functions and traits[1,6,7]. For these reasons, gene duplication is an important evolutionary mechanism for rapid adaptation to novel metabolic and ecological niches[8–12]. Recently duplicated and thus functionally redundant genes often revert to a single-copy state in the absence of selection[13], suggesting that selection is required to maintain duplicated genes. Indeed, selection for strong gene expression is a key factor for the preservation of duplicated antibiotic resistance genes (ARGs) on plasmids[14]. In addition, recent metagenomic studies indicate that

copy number variation in the human microbiome is common and influences human health[15,16].

Laboratory experiments have demonstrated that positive selection can drive the rapid evolution of gene duplications, due to the rapid kinetics of molecular mechanisms like tandem amplifications[4,17]. While several studies have examined tandem duplications and gene amplifications under laboratory selection for drug resistance[3,18–20] or specific metabolic functions[8,9,11], few studies have examined the role of mobile genetic elements (MGEs) in promoting gene duplications.

Following Partridge et al. [21], we define MGEs as "elements that promote intracellular DNA mobility (e.g., from the chromosome to a plasmid or between plasmids) as well as those that enable intercellular

[1]Center for Quantitative Biodesign, Duke University, Durham, NC, USA. [2]Department of Biomedical Engineering, Duke University, Durham, NC, USA. [3]Department of Biostatistics and Bioinformatics, Duke University School of Medicine, Durham, NC, USA. [4]Center for Biomolecular and Tissue Engineering, Duke University, Durham, NC, USA. [5]Department of Molecular Genetics and Microbiology, Duke University School of Medicine, Durham, NC, USA. ✉e-mail: you@duke.edu

DNA mobility". In our experiments, we focus on transposons and plasmids, which are known to mediate the horizontal transfer of ARGs in microbial communities[5,22]. Our bioinformatics analyses more broadly examine genes encoding MGE components, including genes involved in transposon, integrase, bacteriophage, and plasmid functions.

Previously, we showed that antibiotics select for the movement of transposable ARGs from chromosomes onto multicopy plasmids, because the increased copy number of ARGs on multicopy plasmids leads to higher expression of those genes and thus higher resistance[5]. Based on those findings, we reasoned that antibiotic selection would also favor duplications of ARGs, generated by intrachromosomal transposition events. We tested this hypothesis using mathematical modeling, experimental evolution, and genome sequencing to confirm the location and copy number of transposable ARGs in evolved populations.

Based on these experimental findings, we reasoned that antibiotic use should enrich specific populations of bacteria with duplicated ARGs. Several recent studies have reported cases of gene duplications in clinical antibiotic-resistant isolates, using long-read sequencing or qPCR to measure resistance gene copy number[23–33]. However, it is not known whether these cases represent a broader trend. To address this question, we examined the distribution of duplicated genes in tens of thousands of complete bacterial genomes that were sequenced with long-read sequencing technologies.

To date, few studies have systematically examined duplicated genes in bacterial genomes[34], due to the difficulty of resolving identical sequence repeats with second-generation short-read sequencing technologies[35]. Such sequence repeats facilitate gene duplication[2], but also hamper their discovery by short-read sequencing, due to read alignment inaccuracies[36]. These issues also plague genome assembly from complex metagenomic samples[37]. Long-read sequencing is critical because long reads can span repeat regions, including transposons and duplicated genes. This resolves ambiguities in copy number variation, including the coexistence of plasmids, in a given isolate or metagenomic sample[35,38].

Here, by combining modeling, experiments, and bioinformatic analyses, we show that MGEs serve as potent drivers of gene duplications, that gene duplications mediated by MGEs are often adaptive, that duplicated ARGs are enriched in isolates from humans and livestock (the microbial environments most associated with antibiotic use), that duplicated ARGs are further enriched in clinical antibiotic-resistant isolates, and that duplicated ARGs are far more likely to be associated with MGEs than single-copy ARGs. These findings indicate that duplicated genes often encode functions undergoing positive selection and horizontal gene transfer in microbial communities.

## Results

### Antibiotics select for duplicated ARGs

Our basic intuition is that mutants with a duplicated ARG can invade an ancestral clonal population with a single-copy resistance gene, given a sufficiently high concentration of antibiotic. To formalize this idea, we built a mathematical model (Fig. 1A, Supplementary Data 1) based on the framework in our previous study[5]. This model involves three subpopulations of bacteria: the first carries an ARG on the chromosome (Type 1), the second has a duplicated ARG on the chromosome (Type 2), and the third carries a duplicated ARG on a plasmid (Type 3). The ARG confers a fitness benefit in the presence of antibiotics due to resistance, and additional copies confer stronger resistance. However, the additional copies may incur a fitness cost in the absence of antibiotic. We assume that all cells contain a plasmid. By letting the copy number of the plasmid be a free parameter of the model, we can also model the no plasmid case (plasmid copy number = 0). The fitness of each population therefore depends on antibiotic concentration, the cost of ARG expression, and the effective number of ARG copies per cell in each subpopulation, which depends on plasmid copy number (Methods: Mathematical model: Fitness functions).

Under antibiotic selection, one of the subpopulations with the additional ARG copy rapidly outcompetes the others, depending on which has the highest fitness. When the cost of expressing additional ARG copies is low, then the Type 3 subpopulation, which contains duplicated ARGs on the plasmid, dominates (Fig. 1B). When the cost of expressing the ARG on the plasmid outweighs the benefit of resistance, the Type 2 subpopulation, which contains duplicated ARGs on the chromosome, dominates (Supplementary Data 1). By defining a "Duplication Index" as the fraction of the population with a duplicated ARG, we find that duplicated ARGs rapidly establish throughout the population at a threshold antibiotic concentration. As the cost of ARG expression increases, this threshold concentration increases. This is shown by the rightward shift of curves representing higher ARG expression costs in Fig. 1C. In addition, as the transposition rate of the transposable ARG increases, the time for establishment of duplicated ARGs in the population decreases, as shown by a leftward shift of curves representing higher transposition rates in Fig. 1D. Furthermore, the model shows that for any given ARG expression cost, duplicated ARGs will establish in the population when both the transposition rate and antibiotic concentration are sufficiently high (Fig. 1E). Altogether, these results highlight what the dynamics of antibiotic selection and ARG duplication could look like, and illustrate a basic model that can be tested experimentally.

We tested the core prediction of this model– that antibiotics select for duplicated ARGs– by carrying out evolution experiments with *E. coli* strains harboring a minimal transposon composed of a *tetA* tetracycline resistance gene flanked by 19-base-pair terminal repeats. This mini-transposon is mobilized by an external *Tn5* transposase in the chromosome[39]. We carried out 9-day selection experiments with *E. coli* DH5α and sequenced populations resistant to 50 μg/mL tetracycline, varying plasmid, the presence of active transposase, and the basal expression of the *tetA* resistance gene. We also evolved and sequenced a parallel set of control populations that were propagated without tetracycline (Supplementary Data 2). Multiple transpositions of the *tetA-Tn5* transposon to both chromosome and plasmid are observed in the presence of active transposase. In the absence of active transposase, we see parallel mutations affecting the *tetA* promoter and the native efflux pump regulatory genes *robA*, *marR* and *acrR* (Fig. 1F). By contrast, no gene duplications were observed in the no-antibiotic control populations, nor was any parallel evolution observed (Supplementary Data 2). This finding implies that tetracycline treatment selected for the *tetA* duplications and the other resistance mutations observed across replicate populations (Fig. 1F).

Given this finding, we asked whether duplications could arise as a short-term evolutionary response, in a wild-type K-12 MG1655 genetic background. Given the high activity of the synthetic *tetA-Tn5* transposon, one day of tetracycline selection ( ~ 10 bacterial generations) was sufficient to drive duplications of the tetracycline resistance gene to observable allele frequencies across all replicate populations, both in the presence and absence of plasmids (Fig. 2A). By contrast, no duplications were observed in the no-antibiotic control populations (Figure 2A, B, C, D). No *tetA* duplications were observed in the absence of transposase, although gene amplifications of the native *acrAB* antibiotic efflux pump were seen (Fig. 2D). Since no *tetA* duplications or other resistance mutations were observed in the no-antibiotic control treatment (Supplementary Data 2), we infer that tetracycline treatment directly selected for the observed *tetA* duplications, *acrAB* amplifications, and other resistance mutations. We then replaced the *tetA* gene in the minimal *Tn5* transposon with *smR*, *kanR*, *ampR*, and *cmR* genes conferring resistance to spectinomycin, kanamycin, carbenicillin, and chloramphenicol, and repeated our one-day selection experiment using these four antibiotics. ARG duplications were observed in 8 out of 8 evolved populations, across all four antibiotic treatments (Supplementary Fig. 1). Together, the mathematical model and these evolution

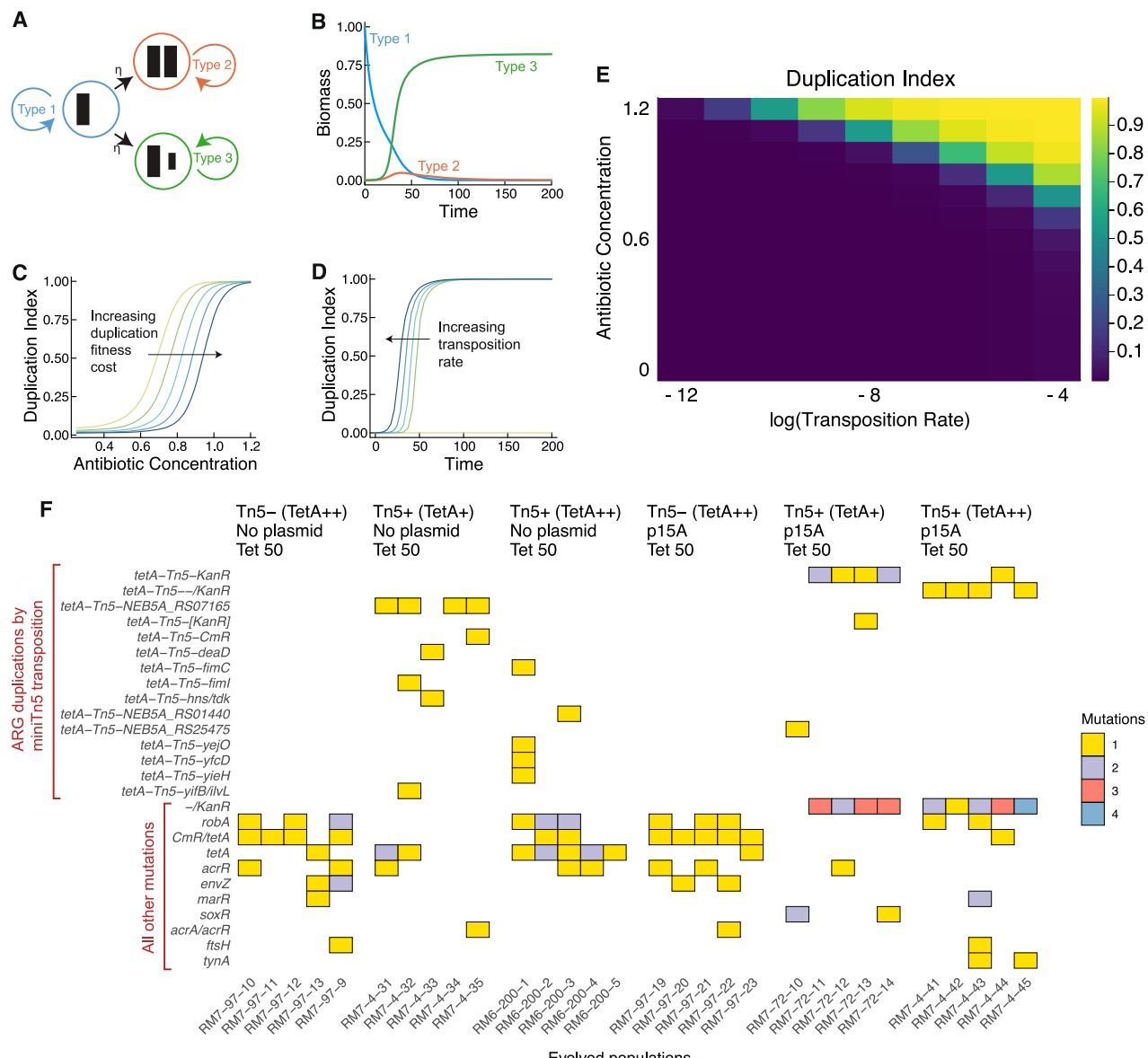

**Fig. 1 | Mathematical modeling and laboratory evolution with *E. coli* K-12 DH5α demonstrate that antibiotic selection is sufficient to drive the rapid evolution of antibiotic resistance through the duplication of antibiotic resistance genes.** Source data are provided in the Source Data File. **A** State diagram for the mathematical model. The three states represent cells with an ARG on the chromosome (Type 1), cells with an additional ARG on the chromosome due to duplication, including transposition-based mechanisms (Type 2), and cells with an ARG on the chromosome and an ARG on its plasmid due to transposition (Type 3). **B** Under sufficiently strong antibiotic selection and with low cost of expression, cells with ARGs on the plasmid dominate the population. The simulation result in this panel uses the following parameter settings (arbitrary units): Antibiotic Concentration $A = 2.0$, Duplication Cost $c = 0.1$, Transposition Rate $\eta = 0.0002$, Dilution Rate $D = 0.1$, Plasmid copy number $y = 2$ (Methods: Mathematical model). Under these conditions, the fitnesses of the three subpopulations are ordered $f_1 < f_2 < f_3$. **C** Cells containing D-ARGs dominate population dynamics at sufficiently high antibiotic concentrations, even if the cost of maintaining the D-ARG varies. Duplication Index is defined as the fraction of cells containing D-ARGs. The simulation result in this panel uses the following parameter settings (arbitrary units): $A = 2.0$, $\eta = 0.0002$, $D = 0.1$, $y = 2$. Colors shift from yellow to blue as the fitness cost of carrying duplicated ARGs increases. The yellow curve represents Duplication Cost $c = 0.05$, and

each successively darker curve represents an increment of 0.05, up to the darkest curve of $c = 0.25$. See Supplementary Data 1 for further details. **D** Increasing the transposition rate reduces the delay until strains with duplicated ARGs take over the population. The simulation result in this panel uses the following parameter settings (arbitrary units): $A = 2.0$, $c = 0.1$, $D = 0.1$, $y = 2$. Colors shift from yellow to blue as the transposition rate $\eta$ increases. $\eta$ is varied on a log-scale from 0, $2 \times 10^{-6}$, $2 \times 10^{-5}$, $2 \times 10^{-4}$. **E** Duplicated ARGs establish in the population when both the transposition rate and antibiotic concentration are sufficiently high. As above, Duplication Index is defined as the fraction of cells containing D-ARGs. The simulation result in this panel uses the following parameter settings (arbitrary units): $c = 0.1$, $D = 0.1$, $y = 2$. Antibiotic concentration $A$ is varied from 0.0 to 1.2 in increments of 0.1, and transposition rate $\eta$ is varied on a $\log_{10}$-scale from $10^{-12}$ to $10^{-4}$. **F** Genome sequencing reveals targets of positive selection after 9 days of growth with increasing tetracycline concentrations up to 50 μg/mL tetracycline. Rows indicate genetic loci, and columns indicate replicate evolved populations. The color of each entry of the matrix represents the number of distinct mutations found at that locus in the population: yellow for one mutation, purple for two, red for three, and blue for four. Mutations involving the *tetA-Tn5* mini-transposon have a *tetA-Tn5-* prefix.

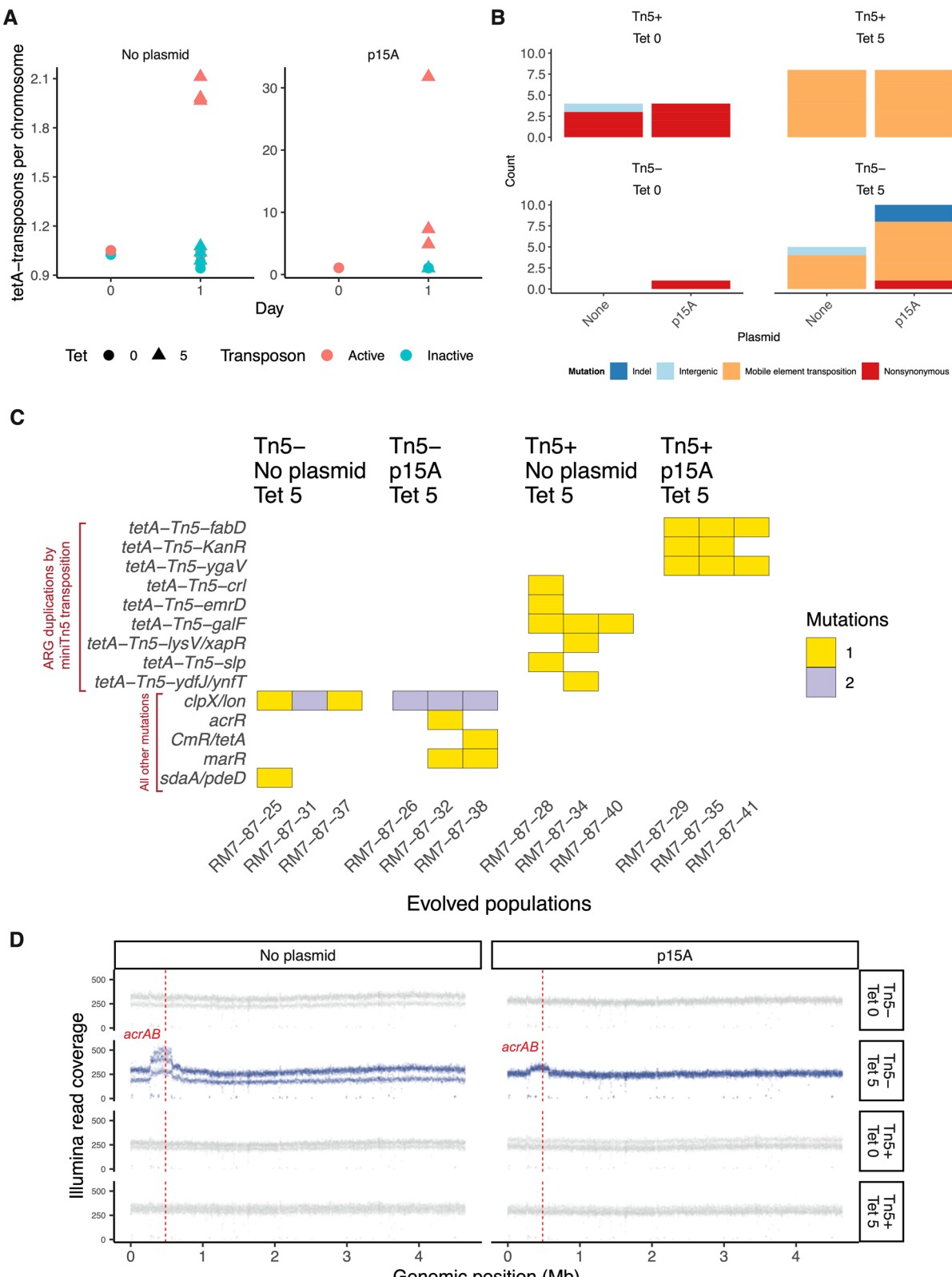

experiments demonstrate the that antibiotic selection can drive the evolution of duplicated ARGs via intragenomic transposition.

**Isolates from humans and livestock have elevated proportions of duplicated ARGs**

To examine the relevance of duplicated ARGs in the ecological context of natural and clinical isolates, we downloaded all complete and fully

annotated bacterial genomes from NCBI RefSeq[40] passing additional quality control checks (25,224 genomes were downloaded and 24,102 genome passed quality control, see Methods: Curation of complete bacterial genomes) and grouped them into 7 different ecological categories (excluding "Unannotated") based on their isolation source and host source metadata (Supplementary Data 3). We used categories similar to, but with higher granularity than, the ProGenomes2

**Fig. 2 | Laboratory evolution with *E. coli* K-12 MG1655 demonstrate that antibiotic selection is sufficient to drive the rapid evolution of antibiotic resistance through the duplication of antibiotic resistance genes.** 12 replicate populations were evolved for one day under tetracycline selection, and another 12 replicate populations were evolved in LB without antibiotic as a control. Each panel shows a result generated by whole-population Illumina sequencing of these evolved populations. See Supplementary Fig. 1 for the results of additional experiments showing generality across antibiotic resistance genes. Source data are provided in the Source Data File. **A** One day of tetracycline selection was sufficient to drive an increase in ARG copy number. No change in *tetA* copy number occurred in the no tetracycline control treatment, or when the transposase was not present. **B** Antibiotic selection enriches for mobile element transpositions, even when the Tn5 transposase is not present. Parallel native mobile element insertions into the

promoter of the *lon* gene encoding the Lon protease, which regulates native efflux pump expression, is the cause (see **C**). **C** Genome sequencing reveals targets of positive selection after 1 day of growth under a treatment of 5 μg/mL tetracycline. Multiple transpositions of the *tetA-Tn5* mini-transposon to both chromosome and plasmid are observed in the presence of active transposase. In the absence of active transposase, we see parallel mobile element insertions into the promoter of *lon*, as well as mutations affecting the native efflux pump regulatory genes *marR* and *acrR*. **D** After 1 day of growth under 5 μg/mL tetracycline, all six of the populations that lack active transposase (shown in blue) show chromosomal amplifications around the location of the native antibiotic resistance efflux pump *acrAB* in the K12 MG1655 NC_000913 reference genome. Populations with Tn5 transposase, or that were not treated with antibiotic, lack these amplifications.

Database[41]. We then examined the distribution of duplicated ARGs across these 7 ecological categories, spanning 18,938 genomes after excluding those that were assigned to the "Unannotated" category (Supplementary Data 3). We define "duplicated" genes based on 100% amino-acid sequence identity. Therefore, our analysis calls a pair of genes within a genome that only differ by silent (synonymous) substitutions "duplicated", while a pair of genes that differ by a single amino-acid change would be called as a pair of "single-copy" genes (Fig. 3).

The 100% sequence identity threshold is critical for defining duplicated genes. When a protein is encoded by two separate loci in the genome, one can assume that its production is redundant. This assumption is much harder to justify if the two copies differ by even a small number of amino acid substitutions, since those may nevertheless have substantial effects on protein function. Given the redundant production of a protein at two or more loci, one can suppose one of two possibilities. Either the duplication event has occurred in the recent past, such that not enough time has passed for the two copies to diverge in sequence, or the production of the protein from multiple loci may be evolving under strong purifying selection, such that the sequence found at multiple loci is being preserved as time passes.

Our operational definition of duplicated genes does not take plasmid copy number into account, such that a protein encoded on a multi-copy plasmid would be classified as "single-copy" if there is no additional sequence encoding the same protein elsewhere in the genome. While modifying our definition such that all plasmid-borne proteins count as "duplicated" does not change our conclusions, it has the disadvantage of collapsing the useful distinction between proteins encoded once or multiple times on a plasmid.

We estimated the proportion of isolates carrying duplicated ARGs in each ecological category: this estimate represents the empirical probability of whether an isolate from a given ecological category has duplicated ARGs. Isolates from humans and livestock show significantly higher proportions of isolates carrying duplicated ARGs, in comparison to the other categories (Fig. 4A and Supplementary Table 1). This trend holds for many different classes of antibiotics, including chloramphenicol, tetracycline, MLS antibiotics, beta-lactams, diaminopyrimidines, sulfonamides, quinolones, aminoglycosides, and macrolides. (Supplementary Fig. 2). By comparison, most isolates in all categories have at least one annotated ARG (Fig. 4B, Supplementary Table 2), and at least one duplicated gene (Fig. 4C and Supplementary Table 3). This result holds for duplicated ARGs found solely on chromosomes or plasmids (Supplementary Fig. 3).

We checked the robustness of the pattern shown in Fig. 4A with further computational controls. We reasoned that the association between duplicated ARGs and isolates from humans and livestock could be affected by both the over-representation of some bacterial taxa, as well as phylogenetic correlations between highly related isolates. To evaluate these possibilities, we compared the number of isolates per bacterial genus to the number of isolates containing duplicated ARGs per bacterial genus. *Klebsiella* and *Escherichia* are

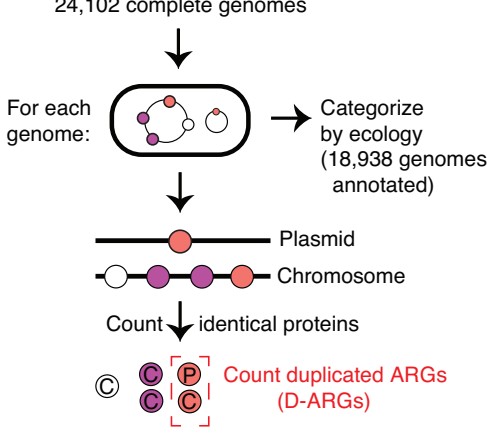

**Fig. 3 | Bioinformatic analysis workflow.** Genes, represented as colored "beads on a string", are grouped together based on 100% protein sequence identity. The location of identical proteins (plasmid, chromosome, or unassembled contig sequence) is recorded, along with the number of copies in those locations. Multiple identical protein sequences in a genome are called "duplicated", while unique protein sequences are called "single-copy". Antibiotic resistance genes were scored based on NCBI RefSeq protein product annotation. Each genome is categorized into one of twelve ecological categories, or as "Unannotated", based on the host and isolation source metadata in its NCBI RefSeq record.

over-represented among both the isolates as well as isolates containing duplicated ARGs. Several other genera containing human commensals and pathogens (*Staphylococcus, Salmonella, Pseudomonas, Acinetobacter*) are highly represented and often have duplicated ARGs (Supplementary Fig. 4A). After removing the bacterial genera that are most enriched with isolates containing duplicated ARGs, the overall difference between categories is much smaller, although isolates from livestock are still most likely to contain duplicated ARGs (Supplementary Fig. 4B). Within the genera that are most enriched with isolates containing duplicated ARGs, isolates from humans and livestock are still much more likely to contain duplicated ARGs (Supplementary Fig. 4C). To examine the effect of phylogenetic correlations between highly related isolates, we downsampled the data in two ways. First, we used Assembly Dereplicator[42] to remove genomes based on a pairwise phylogenetic distance threshold (Mash distance > 0.005). Second, we downsampled the data to one genome per species. After downsampling, isolates from humans and livestock are still most likely to contain duplicated ARGs compared to the other categories (Supplementary Fig. 4D, E). This analysis indicates that the association between duplicated ARGs and isolates from humans and livestock is robust, but most relevant for a small number of bacterial genera. Within those genera, strains isolated from humans and livestock are much more likely to carry duplicated ARGs.

We also examined all the genes, rather than the isolates, in each ecological category. Although genes within a genome have correlated

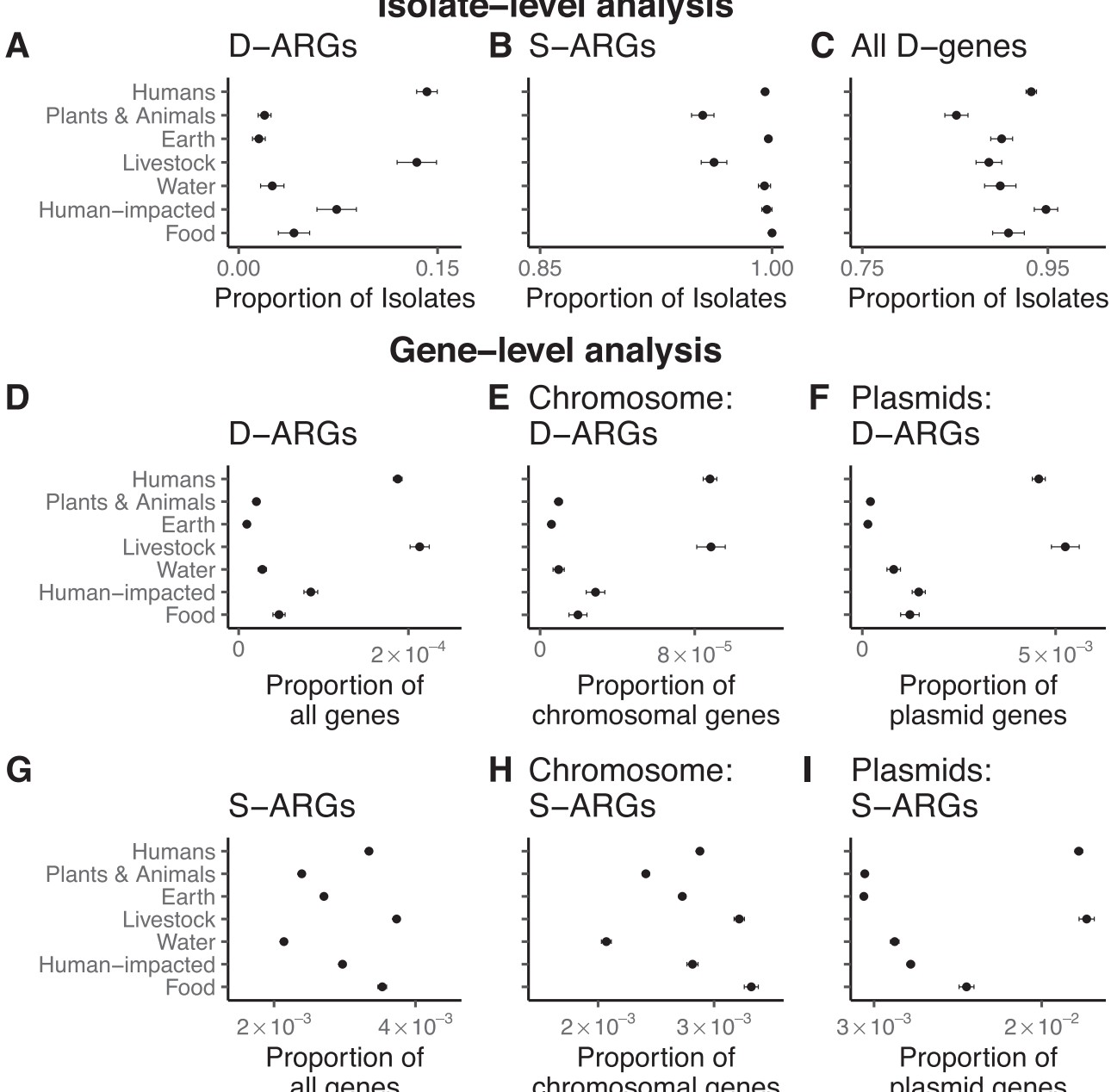

**Fig. 4 | Bacteria isolated from humans and livestock are much more likely to have duplicated antibiotic resistance genes (D-ARGs) compared to bacteria isolated from other environments; furthermore, D-ARGs are enriched on the chromosomes and plasmids of bacteria isolated from humans and livestock.** Error bars are 95% binomial proportion confidence intervals, calculated using the formula $p \pm Z_{\alpha/2}\sqrt{(\frac{p(1-p)}{n})}$, where $p$ is the proportion, $n$ is the sample size, and $Z_{\alpha/2} = 1.96$. The measure of center for the error bars is the proportion $p$ that is relevant for a given figure panel. Numerical reporting, including sample sizes, are listed in Supplementary Tables 1, 2, 3, 4, 5. Source Data are also provided in the Source Data File. **A** D-ARGs are specifically enriched in bacterial isolates from humans and livestock. See Supplementary Table 1 for numerical reporting. **B** The vast majority of isolates contain at least one single-copy antibiotic resistance gene (S-ARG). See Supplementary Table 2 for numerical reporting. **C** The vast majority of isolates contain at least one duplicated gene (D-gene). See Supplementary Table 3

for numerical reporting. **D** D-ARGs represent a higher fraction of genes found in bacteria isolated from humans and livestock compared to bacteria in the other ecological categories. See Supplementary Tables 4 and 5 for numerical reporting. **E** Chromosomal D-ARGs are enriched in bacteria isolated from humans and livestock. See Supplementary Tables 4 and 5 for numerical reporting. **F** Plasmid D-ARGs are enriched in bacteria isolated from humans and livestock. See Supplementary Tables 4 and 5 for numerical reporting. **G** S-ARGs represent a higher fraction of genes found in bacteria isolated from humans and livestock compared to the other ecological categories. See Supplementary Tables 4 and 5 for numerical reporting. **H** Chromosomal S-ARGs are enriched in humans and livestock. See Supplementary Tables 4 and 5 for numerical reporting. **I** Plasmid S-ARGs are enriched in bacteria isolated from humans and livestock. See Supplementary Tables 4 and 5 for numerical reporting.

evolutionary histories due to vertical descent, this analysis provides additional context for our main results, and uses a methodology that is consistent with metagenomic studies that focus on the abundance of genes and their functional annotations, rather than genomes per se, as

ecological markers[43]. Duplicated ARGs encompass a much higher proportion of genes in the human-host and livestock categories in comparison to the other ecological categories (Fig. 4D, Supplementary Table 4). This trend holds for both chromosomal genes (Fig. 4E) as well

as for plasmid genes (Fig. 4F). The gene-level analysis also shows that single-copy ARGs are frequent in the human-host and livestock categories (Fig. 4G, Supplementary Table 5), again for both chromosomal genes (Fig. 4H) and plasmid genes (Fig. 4I). When examining separate classes of antibiotics, we find that single-copy tetracycline and sulfonamide resistance genes are most common in the human-host and the livestock category (Supplementary Fig. 5).

## Clinical antibiotic-resistant isolates are enriched with duplicated ARGs

To validate the medical relevance of these findings, we searched the recent literature for datasets of bacterial genomes satisfying three criteria: (1) High quality, publicly available, and fully annotated genomes sequenced by long-read technologies; (2) known provenance from clinical antibiotic-resistant isolates; and (3) independence from our main dataset of complete genomes from NCBI RefSeq, to rigorously test the hypothesis that antibiotic treatment selects for duplicated ARGs. We found four genomic datasets satisfying these criteria and measured the extent to which each dataset contained duplicated ARGs. First, we re-examined the genomes of 12 clinical extended-spectrum beta-lactam (ESBL) resistant *E. coli* isolates from Duke University Hospital, that were previously sequenced by our group and colleagues[44]. 6 of these 12 isolates contain duplicated ARGs (Supplementary Fig. 6). Second, we also examined the genomes of 46 ESBL-resistant and vancomycin-resistant *Enterobacter*, *Escherichia*, and *Klebsiella* that were sequenced as part of the BARNARDS study of antibiotic resistance at 12 clinical sites in 7 countries across Africa and South Asia[45]. 23 of these 46 isolates contain duplicated ARGs (Supplementary Fig. 7). Third, we examined the genomes of 149 clinical ESBL-like *E. coli* isolates from a tertiary care hospital[46]. 36 of these 149 isolates contain duplicated ARGs (Supplementary Fig. 8). Fourth, we examined the genomes of 114 clinical ESBL-resistant isolates from an Australian ICU[47]. 20 of these 114 isolates contain duplicated ARGs (Supplementary Fig. 9). Altogether, 26% of these clinical antibiotic-resistant isolates (85 out of 321) contain duplicated ARGs. By contrast, 14% of the human isolates in our main dataset (1054 out of 7490) contain duplicated ARGs (Fig. 2A and Supplementary Table 1). Therefore, the clinical antibiotic-resistant isolates in these additional datasets are enriched with duplicated ARGs, relative to the general human isolates in our main dataset (Binomial test: $p < 10^{-8}$).

The clinical genomes isolated from an Australian ICU (NCBI BioProject PRJNA646837) had complete and fully annotated plasmid sequences, so we examined plasmid copy number relative to chromosome across this set of clinical ESBL-resistant strains[47]. Plasmids carrying beta-lactamases had significantly higher copy number than plasmids carrying other kinds of resistance genes (Mann-Whitney $U$-test, $p < 10^{-16}$). However, plasmids carrying ARGs had significantly lower copy numbers than plasmids without ARGs (Mann-Whitney $U$-test, $p < 10^{-16}$). Regardless, these data show that plasmid copy number tends to increase the copy number of linked ARGs (Supplementary Fig. 10).

## Antibiotic resistance genes are associated with plasmids

If ARGs evolve additional copies under selection for increased gene dosage, then we expect that ARGs, especially those associated with MGEs, would often occur on plasmids, because plasmids often have a higher copy number than the chromosome. We tested this prediction by comparing the distribution of single-copy ARGs on chromosomes and plasmids to the distribution of duplicated ARGs on chromosomes and plasmids (Fig. 4E, F, H, I).

Across ecological annotations, most duplicated ARGs occur on plasmids, while duplicated genes overall are more common on chromosomes (Supplementary Table 4). Specifically, 3360 duplicated ARGs occur on chromosomes, while 4289 occur on plasmids, in comparison to 850,342 non-ARG duplicated genes on chromosomes and 119,937

non-ARG duplicated genes on plasmids. By constructing a contingency table with these numbers, we find an overwhelming association between duplicated ARGs and plasmids (Fisher's exact test: $p < 10^{-16}$). Furthermore, duplicated genes encoded solely on plasmids are more likely to encode antibiotic resistance and functions other than those associated with MGEs (40,714 duplicated genes encoding ARGs and other non-MGE functions, compared to 32,008 duplicated genes encoding MGE functions), in comparison to both duplicated genes encoded solely on the chromosome (297,239 duplicated genes encoding ARGs and other non-MGE functions, compared to 451,951 duplicated genes encoding MGE functions), and duplicated genes encoded on plasmids and the chromosome (31,555 duplicated genes encoding ARGs and other non-MGE functions, compared to 124,615 duplicated genes encoding MGE functions), as shown in Supplementary Fig. 11. Therefore, duplicated genes found solely on plasmids have higher proportions of ARGs and other functional genes, in comparison to duplicated genes found solely on chromosomes (Binomial test: $p < 10^{-16}$) and duplicated genes found on both chromosomes and plasmids (Binomial test: $p < 10^{-16}$).

Single-copy ARGs also show strong associations with plasmids (Supplementary Table 5). 189,137 single-copy ARGs occur on chromosomes, while 23,315 occur on plasmids, in comparison to 67,078,963 non-ARG single-copy genes on chromosomes and 1,967,705 non-ARG single-copy genes on plasmids. In this case as well, we find an overwhelming association between single-copy ARGs and plasmids (Fisher's exact test: $p < 10^{-16}$). Therefore, the statistical association between ARGs and plasmids is general. These results also show that in terms of absolute numbers, most single-copy ARGs occur on chromosomes, while most duplicated ARGs occur on plasmids.

## Duplicated genes are more frequently associated with MGEs than single-copy genes

When we examine the functional annotation of duplicated genes (Methods: *Sequence classification based on functional annotation*), we find that ~60% (608,465 out of 977,928 duplicated genes) are associated with MGE components, such as genes involved in transposon, integrase, bacteriophage, and plasmid functions (Fig. 5A). This finding is intuitive, since this class of genes often encode components of "DNA cut-and-paste" and "DNA copy-and-paste" machinery. This trend holds for both duplicated genes found on chromosomes (539,878 out of 853,702 duplicated chromosomal genes) as well as for those found on plasmids (68,587 out of 124,226 duplicated plasmid genes), and this trend holds across all ecological categories. By contrast, less than 5% of single-copy genes on chromosomes encode functions related to MGEs (2,511,319 out of 67,268,100 single-copy chromosomal genes), while ~15% of single-copy genes on plasmids encode MGE-related functions (316,013 out of 1,991,020 single-copy plasmid genes) (Fig. 5B).

## Duplications of ARGs and MGEs reflect non-random evolutionary forces

Suppose no evolutionary forces such as selection, horizontal gene transfer, or associations with MGEs affect the probability that a gene undergoes gene duplication. Under this null hypothesis, the probability that a gene of a given functional class is duplicated should be proportional to the fraction of single-copy genes represented by this functional class. Deviations from this null expectation (i.e., the ratio of the proportion of duplicated genes to the proportion of single-copy genes equals one, implying that the log-ratio equals zero) indicates that the frequency of duplicated genes is being driven away from equilibrium by evolutionary forces. A visual explanation of this method and the null expectation is shown in Supplementary Fig. 12.

Using this method, we find that duplicated ARGs are enriched in bacteria isolated from humans, livestock, water, and human-impacted environments; fit the null expectation for bacteria isolated from food, and are depleted from plants, animals, and earth (Fig. 5C). Duplicated

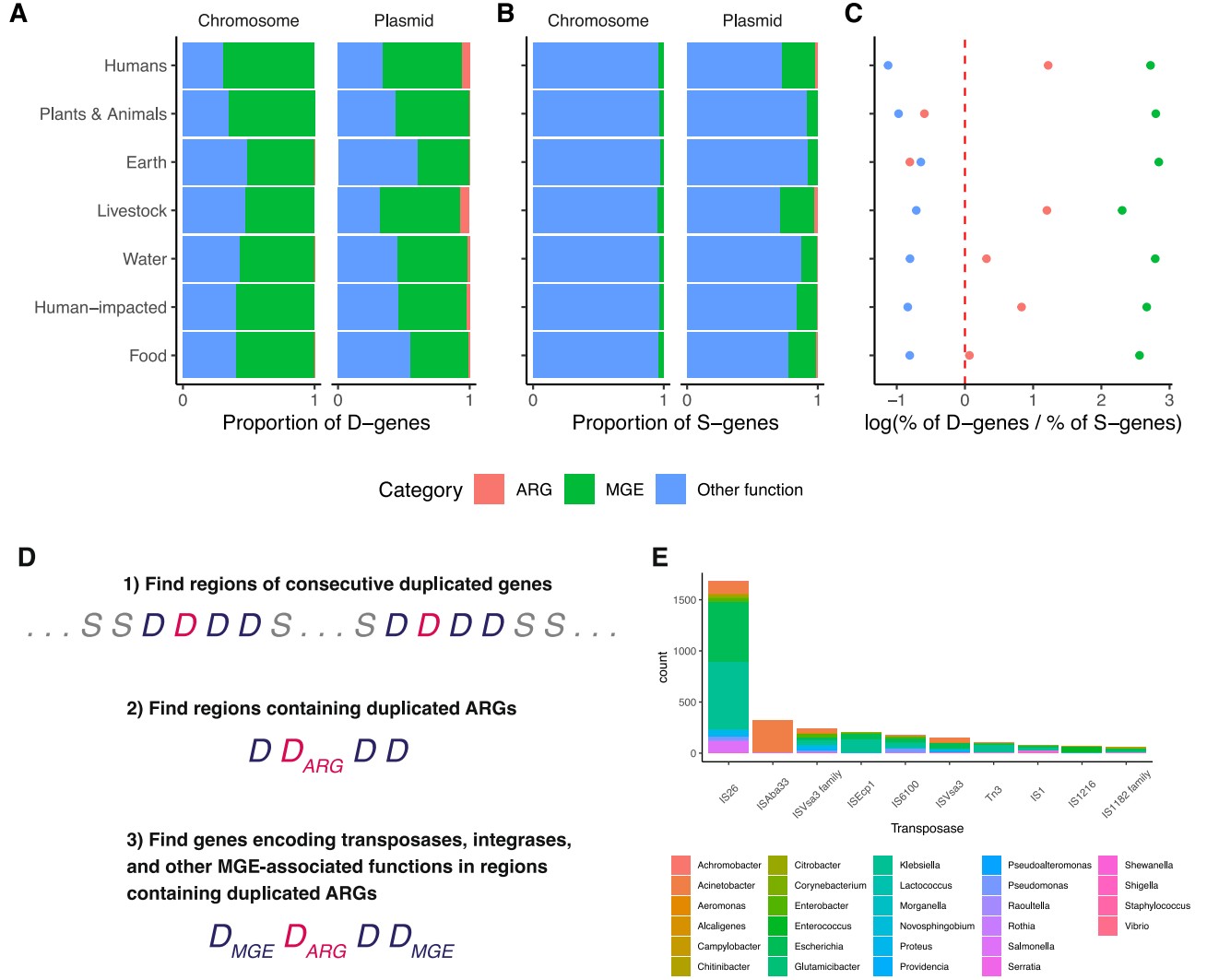

**Fig. 5 | Selection, horizontal gene transfer, and mobile genetic elements shape the ecological distribution of duplicated genes.** Proteins associated with mobile genetic elements (MGEs) are shown in green; proteins encoded by antibiotic resistance genes (ARGs) are in red; and all other proteins are shown in blue. Source data are provided in the Source Data File. **A** Across all ecological categories, ~50% duplicated genes (D-genes) on chromosomes and plasmids are associated with MGEs. **B** MGE-associated proteins account for <10% of single-copy genes (S-genes) on chromosomes, and 5–25% of S-genes on plasmids. **C** Duplicated ARGs (D-ARGs) are enriched in humans and livestock, and are depleted in most other categories, while duplicated genes associated with mobile genetic element functions are enriched in all ecological categories. The red dashed line indicates the null hypothesis. **D** Workflow for finding duplicated transposases that are linked with duplicated ARGs. **E** The ten most frequent transposases associated with ARGs in regions of consecutive duplicated genes. See Supplementary Fig. S14 for the full distribution.

MGE-associated genes are highly enriched across all environments. Furthermore, duplicated genes encoding all other functions are depleted across all environments (Fig. 5C). This test indicates that duplicated ARGs are being driven to higher-than-expected frequencies in bacteria isolated from humans, livestock, water, and human-impacted environments, due to some evolutionary force like selection, horizontal gene transfer, or both.

**Duplicated ARGs are more frequently associated with MGEs than single-copy ARGs**

To investigate linkage between duplicated ARGs and genes encoding MGE functions, we conducted two analyses. First, we asked whether duplicated ARGs had a higher probability of being flanked by MGE-associated genes, in comparison to single-copy ARGs. This was indeed the case. Examining ARGs across all 18,938 genomes, we found that 4651 out of 9836 duplicated ARGs were flanked by MGE-associated genes, while 37,181 out of 278,074 single-copy ARGs were flanked by MGE-associated genes. Therefore, duplicated ARGs are far more likely

than single-copy ARGs to be linked with MGE-associated genes (Binomial test: $p < 10^{-16}$).

Second, we examined regions of consecutive duplicate genes in each of the 18,938 genomes (Fig. 5D). 3356 regions contain duplicated ARGs and duplicated MGE-associated genes, while 2551 regions contain duplicated ARGs but no duplicated MGE-associated genes. Therefore, annotated MGE-associated genes, such as transposases, are an important factor but are not required for ARG duplication. Of these 6087 regions, 237 contain multiple copies of some duplicated ARG. Therefore, segmental duplications account for a relatively small fraction of duplicated regions in these data. We also compared the relative frequency of transposases and phage integrases in the duplicated regions containing ARGs. 8449 genes encode MGE functions within the duplicated regions containing ARGs. Of these, 5541 encoded transposases. By comparison, 1046 encoded integrases. Therefore, transposases make up a large fraction of the duplicated MGE-function genes associated with duplicated ARGs. Among these, the IS26 transposase has particular significance[48] (Fig. 5E and Supplementary Fig. 13). IS26 is

known to play a major role in the spread of diverse ARGs, including associations with antibiotic resistance plasmids found in carbapenem-resistant *Klebsiella pneumoniae*[49–51].

## Discussion

Our modeling demonstrates that ARG duplication could be an effective mechanism for the evolution of antibiotic resistance. Our genomic analyses show that MGEs, such as the transposons and plasmids in our experiments, can serve as a vehicle for the duplication of ARGs. This finding has relevance for natural and clinical populations, as demonstrated by our bioinformatic analyses. Specifically, the distribution of duplicated ARGs in bacterial genomes isolated from different environments is shaped by non-random evolutionary forces, such as antibiotic selection. This evolutionary process is likely facilitated by association with MGEs. Together, these results imply that antibiotic usage not only enriches for resistant subpopulations: it also selects for mutants with a higher capacity for evolutionary innovation through gene duplication, because one gene copy can maintain ancestral function, while additional copies are free to evolve new functions[1,6]. Our results indicate that MGEs have an intrinsic ability to drive evolutionary innovation through their ability to catalyze the duplication and HGT of passenger genes, such as ARGs, carried within the MGE[52,53].

This work implies that gene duplication in bacteria is often linked to horizontal gene transfer, through a common dependence on MGEs. This conclusion contrasts with previous studies that have treated gene duplication and horizontal gene transfer as distinct mechanisms for genetic innovation in bacteria[54,55]. Our work also contrasts with the majority of experimental studies on gene duplications in bacteria[3,4,14,18,19,56], which have focused on tandem amplifications—and not MGE transposition— as a driver of gene duplication in bacteria[5,57]. A key limitation of our study, however, is that we do not directly identify MGEs in our bioinformatic analysis, due to the technical challenge of doing so comprehensively, reliably, and rapidly across all complete bacterial genomes. The development of databases and tools to identify MGEs across the tree of life will allow researchers to measure the extent to which MGEs contribute to the duplication, diversification, and horizontal transfer of genes under positive selection.

The enrichment we observe of duplicated ARGs in humans and livestock is most likely caused by high rates of antimicrobial exposure[58]. Indeed, our analysis of clinical antibiotic-resistant strains strongly supports antibiotic use as a primary driver for the evolution of duplicated ARGs— even though we do not know the resistance phenotypes or antibiotic treatment history for most of the genomes in this study. Future research could examine the quantitative relationship between antibiotic use and the evolution of duplicated ARGs in settings such as hospitals[59] and factory farms[60,61].

Our analysis has several caveats that warrant analysis in future research. First, our mathematical model implicitly assumes that the mutation rate for genomic resistance mutations is small compared to ARG transposition rates— small enough that genomic resistance mutations can be ignored. More work is needed to measure how the relative magnitudes of these rates, and the relative selective benefits of these molecular mechanisms, affects the evolution of antibiotic resistance by ARG duplication. Second, our experiments focused on *E. coli*, and did not examine whether MGEs promote gene duplication across bacterial species. Given our bioinformatics results, we expect our experimental findings to hold across bacteria, but direct experimental tests are needed. For instance, we expect that the transposition rates may depend on idiosyncratic interactions between a given MGE and its host. In this case, it would be interesting to ask whether the prevalence of a given MGE in a particular bacterial species can predict the transposition rate of that MGE in that species, and thus the importance of particular MGEs for spreading clinically relevant resistance in different pathogen species. Third, while we examined several different ARGs in our experiments, it is unclear whether the type of resistance

mechanism encoded by an ARG (e.g., antibiotic degradation, target modification, efflux pump activity) affects the likelihood of resistance evolution by gene duplication. Given the generality of our bioinformatic results across multiple classes of antibiotics (Supplementary Fig. 2), we predict that the specific molecular mechanism of resistance has little impact on ARG duplication dynamics. Indeed, our mathematical model predicts that the dynamics of duplicated ARGs only depends on transposition rates and the balance of fitness benefits and costs of expressing duplicated ARGs, which needs to be tested by future experiments with a broader set of ARGs operating with different mechanisms.

Finally, our results suggest that duplicated genes, especially those encoded on plasmids, may represent a signature of ongoing horizontal gene transfer and adaptation in microbial communities. If so, one could identify genes under ongoing HGT and natural selection by quantifying gene duplication. For instance, it would be both interesting and important to test whether microbial communities in the permafrost of the Arctic tundra show novel genomic patterns of copy number variation in response to climate change[62], and to test whether pathogens and their hosts show characteristic patterns of copy number variation as they coevolve[63]. Our results also suggest that researchers can compress a bacterial genome into a set of dozens of duplicated genes, while maintaining important evolutionary and ecological information about ongoing HGT and selection. Such simple and practical techniques for producing reduced summaries of biological datasets[64] would allow researchers to scale population genomic analyses of microbial communities and their HGT networks[62,65–67] to millions of microbial genomes and plasmids.

## Methods
### Mathematical model

We built a mathematical model, based on the framework used by Lopatkin et al.[68] and by ref. 5, to study how antibiotic usage can select for duplicated ARGs. A diagram of the model is shown in Fig. 1A. This model involves three subpopulations of bacteria: the first carries an ARG on the chromosome (Type 1), the second has a duplicated ARG on the chromosome (Type 2), and the third carries a duplicated ARG on a plasmid (Type 3). We are interested in the dynamics of the three subpopulations due to selection (growth and dilution) and mutation (duplication by transposition dynamics of the ARG).

See Supplementary Data 1 for an interactive Pluto computational notebook of the model written in Julia 1.8. This notebook can be run by installing and running Pluto.jl within Julia 1.8+ (see instructions at: https://plutojl.org/) and then opening the notebook using the Pluto web browser interface. Unless otherwise stated, the simulation results shown in Fig. 1 use the following default parameter settings (arbitrary units): Antibiotic Concentration $A = 2.0$, Duplication Cost $c = 0.1$, Transposition Rate $\eta = 0.0002$, Dilution Rate $D = 0.1$, Plasmid copy number $y = 2$.

**Model assumptions**

**Selection dynamics.** We assume that there is a steady inflow of nutrients and antibiotic, and a steady outflow of depleted media and cells, reflected by a constant dilution rate, $D$. This assumption allows the population to grow continuously at a steady-state population size. We normalize the number of cells by the carrying capacity, such that each state variable represents the percentage of carrying capacity that is taken up by the subpopulation— note that this is *not* the relative frequency of cells in the population, because the total population may be at a steady state that is less than carrying capacity. The growth rate of each subpopulation is modeled by growth functions $f_i > 0$, that we describe in greater detail below.

**Mutation dynamics.** We define a mutation as a transition from one state to another due to gene duplication by transposition. Each

transition occurs at a constant rate η. We assume that transposon excision rates are negligible, such that duplication events leave the original copy unchanged in the chromosome.

These assumptions lead to a system of differential equations of the form:

$$\frac{dx_i}{dt} = f_i x_i (1 - \Sigma x_i) - D x_i + Q_i \tag{1}$$

where the first term reflects logistic growth at rate $f_i$, the second term reflects constant dilution due to a fixed outflow rate, and the third term wraps up all the state transitions (mutation dynamics).

**Model equations.**

$$\frac{dx_1}{dt} = f_1 x_1 (1 - \Sigma x_i) - D x_1 - 2\eta x_i \tag{2}$$

$$\frac{dx_2}{dt} = f_2 x_2 (1 - \Sigma x_i) - D x_2 + \eta x_1 \tag{3}$$

$$\frac{dx_3}{dt} = f_3 x_3 (1 - \Sigma x_i) - D x_3 + \eta x_1 \tag{4}$$

**Fitness functions.**

$$f_i = (1 - c)^x \frac{K_i^n}{K_i^n + A^n} \tag{5}$$

where A is antibiotic concentration, $K_i$ is the concentration of antibiotic that reduces growth by 50%, $n$ is a Hill coefficient, $c$ is the cost of expressing the ARG, and $x$ is the physical number of ARGs in the cell. We assume that the plasmid has a copy number of $y$, with values ranging from 0 to 4. We assume a Hill coefficient $n = 3$. We also assume that $0 < c < 1$, and that $A \geq 0$. $K$ varies depending on the configuration of ARGs on chromosome or plasmid in each of the three subpopulation types:

$$f_1 = (1 - c) \frac{1^3}{1^3 + A^3} \tag{6}$$

$$f_2 = (1 - c)^2 \frac{2^3}{2^3 + A^3} \tag{7}$$

$$f_3 = (1 - c)^{(1+y)} \frac{(1+y)^3}{(1+y)^3 + A^3} \tag{8}$$

**Strain construction for evolution experiments**
All plasmids used in this study are listed in Supplementary Table 4 of ref. 5 Plasmids were transformed into strains using electroporation or chemical transformation using TSS buffer[69]. Following ref. 5, the helper plasmid pA004 was transformed into *E. coli* DH5α and *E. coli* K-12 MG1655. *tetA*-Tn5 mini-transposons were integrated into the host chromosome as follows. The mini-transposon plasmids contain the HK022 attP sequence. These plasmids were transformed into the host strain containing pA004 and inserted into the HK022 attB site on the *E. coli* chromosome by attB/P recombination. Strains containing integrated mini-transposons were also transformed with the P15A origin plasmid pA031 to examine transposon-plasmid dynamics during experimental evolution.

**Nine-day evolution experiment with E. coli DH5α**
**Culture conditions.** 3 mL cultures were grown in 16 mL 17 x 100 mm culture tubes at 37 C in a 225-rpm shaking incubator. The cultures were propagated by 1:1000 daily serial dilution: 3 μL of the Lysogeny Broth (LB) overnight cultures was used to inoculate 3 mL LB + tetracycline and 3 mL LB without tetracycline as a control.

**Ancestral strains.** The following *tetA*-Tn5 mini-transposons were integrated into *E. coli* DH5α. pB030 contains active Tn5 transposase outside of a mini-transposon containing *tetA* expressed under the strong J23104 promoter (Tn5+ *tetA* ++). pB059 contains a mini-transposon containing *tetA* expressed under the strong J23104 promoter, but does not contain Tn5 transposase (Tn5− *tetA* ++). pB020 contains active Tn5 transposase outside of a mini-transposon containing *tetA* expressed under the weak J23113 promoter (Tn5+ *tetA* +). The P15A plasmid pA031 was also transformed into these three strains. Altogether, six DH5α strains were evolved in LB with increasing tetracycline concentrations over time. These strains varied the presence of active transposase, the presence of an intracellular plasmid, and the strength of the promoter driving *tetA* expression.

**Evolution experiment.** 3 mL LB cultures were inoculated from glycerol stocks of the ancestral clones. The next day, 5× replicate populations were inoculated using 3 μL of the overnight culture for each ancestral clone. Evolving populations were transferred nine times, increasing tetracycline concentrations in LB on each transfer (2, 4, 6, 8, 10, 20, 30, 40, 50 μg/mL). In total, 30 populations evolved in LB + tetracycline from the six ancestral strains. Another 30 populations were evolved in LB without tetracycline as a control. At the end of the experiment, 750 μL of each evolved population was mixed with 750 μL 50% glycerol in 2 mL cryovials and stored at −80 C.

**One-day evolution experiment with E. coli K-12 MG1655**
**Culture conditions.** 1 mL cultures were grown in 2 mL deep 96-well plates covered with a gas-permeable membrane at 37 C in a 700-rpm shaking incubator. The cultures were propagated in one 1:1000 dilution: 1 μL of LB overnight cultures of the ancestral strains were used to inoculate 1 mL LB + 5 μg/mL tetracycline and 1 mL LB without tetracycline as a control.

**Ancestral strains.** The following *tetA*-Tn5 mini-transposons were integrated into *E. coli* K-12 MG1655. pB030 contains active Tn5 transposase outside of a mini-transposon containing *tetA* expressed under the strong J23104 promoter (Tn5+). pB059 contains a mini-transposon containing *tetA* expressed under the strong J23104 promoter, but does not contain Tn5 transposase (Tn5−). The P15A plasmid pA031 was also transformed into these two strains. Altogether, four K-12 strains were evolved overnight in LB + 5μg/mL tetracycline. These strains varied in the presence of active transposase and the presence of an intracellular plasmid.

**Evolution experiment.** A 3 mL LB culture were inoculated from a glycerol stock of an ancestral clone. The next day, 3× replicate populations were inoculated using 1μL of the overnight culture. In total, 12 populations evolved in LB + tetracycline from the four ancestral strains. Another 12 populations were evolved in LB without tetracycline as a control. At the end of the experiment, 750 μL of each evolved population was mixed with 750 μL 50% glycerol in 2 mL cryovials and stored at −80 C.

**Antibiotic generality one-day evolution experiment with E. coli K-12 MG1655**
**Culture conditions.** 1 mL cultures were grown in 2 mL deep 96-well plates covered with a gas-permeable membrane at 37 C in a 700-rpm shaking incubator. The cultures were propagated in one 1:1000 dilution: 1 μL of LB overnight cultures of the ancestral strains were used to inoculate 1 mL LB + 250 μg/mL spectinomycin, 1 mL LB + 250 μg/mL

kanamycin, 1 mL LB + 2000 µg/mL carbenicillin, 1 mL LB + 70 µg/mL chloramphenicol and 1 mL LB without antibiotic as a control.

**Ancestral strains.** The following Tn5 mini-transposons were integrated into *E. coli* K-12 MG1655. pBO90 contains active Tn5 transposase outside of a mini-transposon containing *smR* expressed under the J23109 promoter. pBO91 contains active Tn5 transposase outside of a mini-transposon containing *kanR* expressed under the J23109 promoter. pBO92 contains active Tn5 transposase outside of a mini-transposon containing *ampR* expressed under the J23109 promoter. pBO95 contains active Tn5 transposase outside of a mini-transposon containing *cmR* expressed under the J23109 promoter. Altogether, four K-12 strains were evolved overnight in LB + antibiotic. These strains only varied in the ARG found in the Tn5 mini-transposon integrated into their chromosome.

**Evolution experiment.** A 3 mL LB culture were inoculated from a glycerol stock of an ancestral clone. The next day, 2× replicate populations were inoculated using 1 µL of the overnight culture. In total, 8 populations evolved in LB + antibiotic from the four ancestral strains. At the end of the experiment, 750 µL of each evolved population was mixed with 750 µL 50% glycerol in 2 mL cryovials and stored at −80 C.

## Genomic analysis of evolution experiments

Genomic DNA (gDNA) from ancestral clones and whole-population samples of the evolution experiments was extracted using the GenElute Bacterial Genomic DNA Kit (Sigma-Aldrich). gDNA was sent to SeqCenter (Pittsburgh, PA) for Illumina short-read genome sequencing. Variants were called using *breseq* version 0.37[70] in polymorphism mode, using the following command-line flags: --polymorphism-minimum-variant-coverage-each-strand 4 -b 30 --maximum-read-mismatches 5. For the nine-day evolution experiment, a shell script called *assemble-DH5a-genomes.sh* was used to automate breseq runs and sequence data processing. Analysis of variants across evolved populations was conducted with an R 4.0 script called *DH5a-expt-metagenomics.R*. For the one-day evolution experiment, a shell script called *assemble-one-day-expt-genomes.sh* was used to automate breseq runs and sequence data processing. Variant data was then processed using a Python 3.6 script called *process-one-day-expt-gdiffs.py*. Mini-transposon sequencing coverage data (for estimating copy-number change) was processed using the Python 3.6 script *get-one-day-expt-transposon-coverage.py*. Evidence for chromosomal copy-number changes was processed using an R 4.0 script called *one-day-expt-copy-number-analysis.R*. Analysis of variants across evolved populations was conducted with an R 4.0 script called *one-day-expt-metagenomics.R*. For the one-day antibiotic generality experiment, a shell script called *assemble-generality-expt-genomes.sh* was used to automate breseq runs and sequence data processing. Variant data was then processed using a Python 3.6 script called *process-generality-expt-gdiffs.py*. Mini-transposon sequencing coverage data (for estimating copy-number change) was processed using the Python 3.6 script *get-generality-expts-transposon-coverage.py*. Evidence for chromosomal copy-number changes was processed using an R 4.0 script called *one-day-expt-copy-number-analysis.R*. Analysis of variants across evolved populations was conducted with an R 4.0 script called *antibiotic-generality-expt-analysis.R*.

## Curation of complete bacterial genomes

A list of complete prokaryote genomes was downloaded from: https://ftp.ncbi.nlm.nih.gov/genomes/GENOME_REPORTS/prokaryotes.txt. The list of prokaryote genomes was then filtered for complete genomes, using a Python 3.6 script called *filter-genome-reports.py*. Each bacterial genome contains at least one chromosome and may contain plasmids. A total of 25,224 genomes were downloaded. Assembly statistics for each genome was downloaded using a Python 3.6 script

called *fetch-assembly-stats.py*. The quality of each genome assembly was checked using a Python 3.6 script called *run-QC-and-make-assembly-stats-table.py*, which makes a table of complete genomes that passed additional genome assembly quality control checks for deposition into the NCBI Refseq database. Genomes were further checked for completeness based on their assembly statistics report. Any genomes with gaps or unplaced scaffolds were omitted from the analysis, and all plasmids were checked to ensure the presence of a corresponding chromosome in its genome. Finally, the length of all plasmids and chromosomes was measured with a Python 3.6 script called *count-proteins-and-replicon-lengths.py*, and genomes containing any plasmids larger than its chromosome were tabulated with a Python 3.6 script called *find-bad-replicons.py*. This uncovered 12 genomes containing plasmids larger than the annotated chromosome; these genomes were omitted from the analysis. A total of 24,102 genomes passed these quality control checks. Nucleotide and protein-coding sequences and genome annotation for each of these complete bacterial genomes containing plasmids was then downloaded using two Python 3.6 scripts called *fetch-genome-and-plasmid-cds.py* and *fetch-gbk-annotation.py*. Once the genomes were downloaded, tables summarizing the sequence accessions per genome and the genome annotation metadata were generated by Python 3.6 scripts called *make-chromosome-plasmid-table.py* and *make-gbk-annotation-table.py*. Finally, all plasmids were annotated using MOB-typer 3.1.7[71]. These annotations are provided in Supplementary Data 5.

We used the "host" and "isolation_source" fields in the RefSeq annotation for each genome to place each into the following categories: Marine, Freshwater, Human-impacted (environments), Livestock (domesticated animals), Agriculture (domesticated plants), Food, Humans, Plants, Animals (non-domesticated animals, also including invertebrates, fungi and single-cell eukaryotes), Soil, Sediment (including mud), Terrestrial (non-soil, non-sediment, including environments with extreme salinity, aridity, acidity, or alkalinity), NA (no annotation).

These annotations were based on the annotation categories in the ProGenomes2 Database (Aquatic, Disease associated, Food associated, Freshwater, Host associated, Host plant associated, Sediment mud, Soil). The main difference between our annotation categories and those used in the ProGenomes2 database is that our annotations split host-associated categories based on domestication, and bin all human-host associated microbes together, regardless of disease association. For reproducibility, our annotations are generated using a Python 3.6 script called *annotate-ecological-category.py* and checked for internal consistency using a Python 3.6 script called *check-ecological-annotation.py*. Of the 24,102 genomes passing quality control, 18,938 had sufficient metadata for ecological annotation. This set of 18,938 genomes was used in our data analysis.

Since our results focus on the difference between isolates from human and livestock compared to all other categories, we simplified the data presentation by grouping annotations into larger categories. See the *"Statistical analysis"* section below for details.

## ESBL Escherichia coli isolates from Duke University Hospital

Genome assemblies for the EBSL isolates came from NCBI BioProject PRJNA290784. Only assemblies with long-read PacBio data were examined. Protein-coding sequences in these genomes were tabulated for downstream processing using a Python 3.6 script called *tabulate-proteins-in-clinical-genomes.py*.

## ESBL isolates from the BARNARDS study (Carvalho et al.[45])

Genome assemblies for vancomycin- and ESBL-resistant isolates came from NCBI BioProject PRJNA767644. Only assemblies with long-read Oxford Nanopore data were examined. Protein-coding sequences in these genomes were tabulated for downstream processing using a Python 3.6 script called *tabulate-proteins-in-clinical-genomes.py*.

### ESBL isolates from Barnes-Jewish Hospital (Mahmud et al.[46])

Genome assemblies for ESBL-like resistant isolates came from NCBI BioProject PRJNA824420. All assemblies in this BioProject used long-read Oxford Nanopore data. Protein-coding sequences in these genomes were tabulated for downstream processing using a Python 3.6 script called *tabulate-proteins-in-clinical-genomes.py*.

### Antimicrobial-resistant isolates from an Australian ICU (Hawkey et al.[47])

Genome assemblies for antimicrobial-resistant isolates came from NCBI BioProject PRJNA646837. All complete assemblies deposited in RefSeq (114 genomes) were analyzed. Protein-coding sequences in these genomes were tabulated for downstream processing using a Python 3.6 script called *tabulate-proteins-in-clinical-genomes.py*. To ensure independence from the main set of complete genomes from RefSeq, a Python 3.6 script called *cross-check-Hawkey2022-accessions.py* was used to find matching accessions in the main dataset. These genomes were omitted from the bioinformatic analysis of complete genomes from RefSeq. Given the high-quality plasmid sequences in these data, we estimated plasmid copy number (relative to chromosome) in each of these genomes with a Python 3.6 script called *plasmid-copy-number-analysis.py*. In brief, this script calculates mean short-read sequencing coverage depth per plasmid, normalized by the mean short-read sequencing coverage depth per chromosome, to estimate plasmid copy number relative to chromosome.

### Generation of input files for statistical analysis

A schematic of the basic analysis procedure is shown in Fig. 3. Briefly, the downloaded and annotated bacterial genomes was prepared for analysis in two steps.

A. For every genome, group genes into bins by sequence identity, using a hash table data structure, using the protein-coding sequence as a key (This data structure is also known as a "dictionary").

B. tally the number of sequences on chromosomes, the number of sequences on plasmids, and total number of sequences per genome.

Each protein-coding sequence in each genome, and the number of identical sequences found in each genome, and the location of that sequence (chromosome or plasmid, or scaffold) was tabulated using a set of Python 3.6 scripts. The number of protein-coding genes in each NCBI Nucleotide Accession (each corresponds to a chromosome, plasmid, or scaffold) was tabulated using a Python 3.6 script called *count-cds.py*. All protein-coding sequences across all genomes− including duplicated sequences− were tabulated using a Python 3.6 script called *tabulate-proteins.py*. The number of duplicated and single-copy ARGs adjacent to genes encoding MGE functions were tabulated using a Python 3.6 script called *count-ARG-MGE-adjacencies.py*. Genomic regions consisting of runs of duplicated genes (i.e., each gene within the region is found multiple times somewhere in the genome) were tabulated using a Python 3.6 script called *join-duplications.py*. A set of downsampled genomes based on Mash distance > 0.005 was generated using Assembly Dereplicator v0.3.1 (https://github.com/rrwick/Assembly-Dereplicator), using a Python 3.6 script called *fetch-and-dereplicate-seqs.py*.

### Statistical analysis

All statistical analysis and data visualizations were generated using an R 4.0 script called *ARG-duplication-analysis.R*. To simplify the data presentation, we merged categories as follows. Marine and Freshwater categories were grouped as "Water". Sediment, Soil, and Terrestrial categories were grouped as "Earth". Plant, Agriculture, and Animal categories were grouped as "Plants & Animals". We estimated the proportion of isolates containing particular classes of genes (e.g., duplicated ARGs) within each ecological category, and calculated 95% binomial proportion confidence intervals around the mean, using the formula $p \pm Z_{\alpha/2}\sqrt{(\frac{p(1-p)}{n})}$, where $p$ is the proportion, $n$ is the sample size, and $Z_{\alpha/2} = 1.96$. We also estimated the proportions of particular classes of genes (e.g., duplicated ARGs) compared to all genes within each ecological category, and calculated 95% confidence intervals around the mean using the same method. We further restricted this comparison to just those genes on plasmids as well as just those on chromosomes. We constructed $2 \times 2$ contingency tables to measure associations between ARGs and plasmids or chromosomes, using Fisher's exact test to calculate significance. These basic procedures were then extended to the other specific classes of genes (single-copy ARGs, duplicated ARGs on chromosomes, etc.) reported in the Results. A binomial test was used to compare the probability of a duplicated ARG occurring next to an MGE-associated gene to the probability of a single-copy ARG occurring next to an MGE-associated gene.

### Sequence classification based on functional annotation

Following the method used by ref. 16, the function listed in the protein product field of each sequence's RefSeq annotation was used to classify sequences, using regular expressions.

For genes associated with MGEs we used the following regular expression in R:

"IS|transpos\\S*|insertion|Tra[A-Z]|Tra[0-9]|tra[A-Z]|conjugate transposon|Transpos\\S*|Tn[0-9]|tranposase|Tnp|Ins|ins|relax\\S*|conjug\\S*| mob\\S*|plasmid|type IV|chromosome partitioning|chromosome segregation|Mob\\S*|Plasmid|Rep|Conjug\\S*|capsid|phage|Tail|tail|head|tape measure|antiterminatio|Phage|virus|Baseplate|baseplate|coat|entry exclusion|Integrase|integrase|excision\\S*|exonuclease|recomb|toxin| restrict\\S*|resolv\\S*|topoisomerase|reverse transcrip|intron|antitoxin| toxin|Toxin|Reverse transcriptase|hok|Hok|competence|addiction".

For genes encoding antibiotic resistance, we used the following regular expression in R:

"chloramphenicol|Chloramphenicol|tetracycline efflux|Tetracycline efflux|TetA|Tet(A)|tetA|tetracycline-inactivating|macrolide|lincosamide|streptogramin|Multidrug resistance|multidrug resistance|antibiotic resistance|lactamase|LACTAMASE|beta-lactam|oxacillinase| carbenicillinase|betalactam\\S*|glycopeptide resistance|VanZ|vancomycin resistance|VanA|VanY|VanX|VanH|streptothricin N-acetyltransferase|bacitracin|polymyxin B|phosphoethanolamine transferase| phosphoethanolamine--lipid A transferase|trimethoprim|dihydrofolate reductase|dihydropteroate synthase|sulfonamide|Sul1|sul1|sulphonamide|quinolone|Quinolone|oxacin|qnr|Qnr|Aminoglycoside| aminoglycoside|streptomycin|Streptomycin|kanamycin|Kanamycin| tobramycin|Tobramycin|gentamicin|Gentamicin|neomycin|Neomycin|16 S rRNA (guanine(1405)-N(7))-methyltransferase|23 S rRNA (adenine(2058)-N(6))-methyltransferase|spectinomycin 9-O-adenylyltransferase|Spectinomycin 9-O-adenylyltransferase|Rmt| macrolide|ketolide|Azithromycin|azithromycin|Clarithromycin|clarithromycin|Erythromycin|erythromycin|Erm|EmtA|QacE|Quaternary ammonium|quaternary ammonium|Quarternary ammonium|quartenary ammonium|fosfomycin|ribosomal protection|rifampin ADP-ribosyl|azole resistance|antimicrob\\S*".

We validated the performance of these regular expressions by measuring the precision and recall of these regular expressions in recovering duplicated ARGs and MGE-associated genes found by homology to the Comprehensive Antibiotic Resistance Database (CARD)[72] and the MobileOG-db database[73] of genes associated with MGEs (details in next section). The precision and recall of the ARG regular expression were 0.931 and 0.972, respectively. The precision and recall of the MGE regular expression were 0.786 and 0.871, respectively.

### Sequence classification based on CARD and MobileOG-DB databases

To assess the robustness of our findings, we used the CARD database of ARGs[72] and the MobileOG-db database of genes associated with MGEs[73]

to annotate ARGs and MGE-associated genes in our main dataset of genomes from RefSeq. We used a Python 3.6 script called *protein_csv_to_fasta.py* to generate a FASTA file of all proteins found in the RefSeq genomes, to query CARD and MobileOG-db for homology. We then used a Python 3.6 script called *search-CARD-and-mobileOG-db.py* to find all homologs with >80% identity over >85% of the target sequences in CARD and mobileOG-db, following the protocol used by Gibson et al. [74] to annotate ARGs. The results were then formatted for downstream analysis using a Python 3.6 script called *parse-DIAMOND-results.py*. We used the resulting dataset of annotated ARGs and MGE-associated genes in two ways. First, we used these data as a "ground truth" dataset to measure the precision and recall of our regular expressions in finding ARGs and MGE-associated genes. Second, we used this dataset of annotated ARGs and MGE-associated genes to check that our findings hold when using CARD and mobileOG-db to annotate ARGs and MGE-associated genes (Supplementary Figs. 14, 15, 16).

### Analysis of regions of consecutive duplicated genes

Regions of consecutive duplicated genes in all genomes were found using a Python 3.6 script called *join-duplications.py*. In this analysis, each gene in each genome is scored as being in one of two states: either within a duplicated region, or outside a duplicated region. Since another analysis script, called *tabulate-proteins.py*, makes a table of duplicated genes for each genome, we can use this information to find contiguous regions of duplicated genes in one additional pass through all genomes. Associations between ARGs and genes with MGE-associated functions in these regions were then analyzed in the R 4.0 script *ARG-duplication-analysis.R*. Duplicated transposases associated with duplicated ARGs were clustered based on 99% sequence identity using the Julia 1.8 script *cluster-transposases.jl*. The most common transposases associated with ARGs in regions of consecutive duplicated genes were manually annotated, using BLAST to find the most significant match for these transposases in the ISFinder database[75].

### Reporting summary

Further information on research design is available in the Nature Portfolio Reporting Summary linked to this article.

## Data availability

Source data are provided as a Source Data File. Accessions for the 18,938 complete bacterial genomes from NCBI RefSeq analyzed in this work are listed in Supplementary Data 3. Accessions for the 321 antibiotic-resistant clinical-isolate bacterial complete genomes from NCBI BioProjects PRJNA290784, PRJNA767644, PRJNA824420, and PRJNA646837 analyzed in this work are listed in the legends of Supplementary Fig. 6, 7, 8, and 9. A minimum dataset necessary to interpret, verify and extend the research in this article is available from Zenodo (https://doi.org/10.5281/zenodo.10431250). A 500GB tarball containing all data, source code, and results (1TB uncompressed) is available for download from the You Lab Data Archive at: https://drive.google.com/file/d/1-_pD1G6cQx0KSdUpkNIR-0SGj7EvDS9y/view?usp=sharing. Source data are provided with this paper.

## Code availability

A Github repository containing all data and code sufficient to reproduce the bioinformatic analysis, including statistics and figures, is available at https://github.com/youlab/youlab-ARG-duplications (https://doi.org/10.5281/zenodo.10431250) and at https://github.com/rohanmaddamsetti/youlab-ARG-duplications (https://doi.org/10.5281/zenodo.10431254). This repository also contains all computer codes used to conduct the mathematical modeling and genomic analysis of the 9-day and 1-day evolution experiments. A github repository that only contains these computer codes can be found at: https://github.com/rohanmaddamsetti/AR-gene-plasmid-analysis.

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

## Acknowledgements

We thank the members of the You lab for helpful discussions and comments, and Duke Research Computing for technical assistance and computing resources. This work is partially supported by the National Institutes of Health (L.Y., R01AI125604, R01GM098642, and R01EB031869). The funders had no role in study design, data collection and analysis, decision to publish, or preparation of the manuscript.

## Author contributions

RM conceived the project. RM, YY and LY designed research. RM and TW performed mathematical modeling with guidance from LY. RM conducted experiments. RM, JG, VTH, GSH and HS conducted bioinformatic analyses. RM wrote the manuscript. RM, YY, TW, JG and LY edited the manuscript. LY supervised research.

## Competing interests

The authors declare no competing interests.
