## [Peer Review File · Nature Communications]

Reviewers' Comments:

Reviewer #1:

Remarks to the Author:

The topic is of interest and is timely as AMR is important, and gene duplication is an interesting and poorly studied phenomena because of technical issues. Using new technology of long sequencing reads is a potential solution to support studies of AMR duplication importance.

The abstract could be enhanced by including a statement that the study is multidisciplinary and uses long sequencing reads, experimental data and mathematical modelling.

Overall I found the manuscript interesting. However, I struggled to interpret the figures as they were missing key information. Importantly, the rationale of study design was often not clear. And this meant that most of the main conclusions were not supported in the current format.

Specifically -

1. Mathematical modelling can be useful here, although I couldn't seem to find the assumptions that were used in Figure 1B. The results are highly dependent on the assumptions made. What is the strong antibiotic selection and high cost of expression that is plotted here? Line 89-90 does not seem to be proven by Figure 1B?

2. Figures 1C and 1D, Line 90-93 is dependent on the cost of the ARG expression. What is the evidence for this, or assumptions made when plotting the graphs?

3. Line 94. What is the evidence that antibiotic selection CAUSES high frequency duplication? Presumably it is just selected for?

4. Figure 1E. This figure is very difficult to interpret. What do all the listings on the left mean? Are they mutations? Duplications? What does the x-axis mean? How many replicates? The method says 30 times, but only four presented? How was the data captured or interpreted as it represents a population rather than individual cells? Figure 2C is also similarly difficult to interpret.

5. Figure 2A. Is active meaning the transposon is active? How many replicates were performed, its not possible to tell from the figure. Line 108, what is the evidence for "high frequencies"?

6. Figure 2B. Which method was used to generate the data? What does the y-axis mean? How many replicates?

7. Figure 2C. See point 4.

8. Where is the *acr* locus located? What do the colours mean? How many replicates? Line 110-112, what is the evidence this region is *acr* and was it confirmed by any other method/evidence?

9. Line 105-114, I cannot see the convincing evidence for these statements in Figure 2. "...drive the evolution of..." seems an overstatement when a single tube with antibiotic resistant bacteria is exposed to antibiotic, its not clear that duplication of genes is required.

10. Line 117. The bioinformatics study is potentially interesting. Why were these datasets chosen? Where did they come from? Are they long reads? It would be helpful to include estimates of the number of samples compared in each category in the main text/figures.

11. Figure 4A. "specifically associated" is not an accurate description of the data.

12. Line 157-175 is dependent on Figure S3. I think the data shows that known human pathogens that contain more D-ARGs are found in humans and animals, but its difficult to justify the extensive discussion here is robust. Downsampling is figure S3D not C.

13. Line 192. Why were these isolates chosen? Are they long reads? Without knowing why they were chosen, the assumptions and conclusions made in this paragraph are invalid.

14. Line 223-226, some statistics are needed for this statement in Figure S8.

15. Line 238-244 and Figure 5A, 5B. This does not seem to be intuitive. Are there numbers to support these conclusions instead of just proportions? Why would there be so many duplicate MGE genes found on the chromosome?

16. Line 290. The opening sentence is not really justified by the data. Line 297 is not proven either. Duplication is not the same as location next to an MGE gene.

17. Could the authors differentiate between duplication of an MGE and duplication of an ARG? How often do they both occur together? Or singly? This may help interpret their conclusions? Is it suggested that MGE duplication rather than ARG duplication is the major mechanism of duplication?

Reviewer #2:

Remarks to the Author:

In this study, Maddamsetti and colleagues investigate the relationship between horizontal gene transfer and gene duplication as a driver of the spread of antimicrobial resistance (AMR). This well-conducted study is novel and has multiple sequential components including mathematical modelling, in vitro lab experiments and analysis of a large genomic dataset that together provide an interesting hypothesis for a previously overlooked aspect of AMR spread.

The major issues with the study are:

- The authors conduct in vitro experiments in one species (*E. coli*) and one resistance determinant (*tetA*) and use this to confirm their mathematical modelling findings. I am not sure that this is sufficient evidence to support their claims. It may be worthwhile considering further experiments in other relevant human pathogens with significant burdens of AMR (eg. *Klebsiella pneumoniae*) and more clinically relevant resistance determinants (eg. beta-lactamases).
- The authors recognize that an inability to detect plasmid copy number variation is a limitation of their analyses of publicly available genome data. However, the authors have access to three clinical datasets which have both genome assemblies and read datasets available, as well as potentially numerous other public datasets) (eg. <https://www.ncbi.nlm.nih.gov/pmc/articles/PMC9396894/>). This should make it possible to provide some estimates of the role of plasmid copy number variation.
- The authors imply throughout the study that isolates from human and livestock carry an increased number of duplicated ARGs due to the higher rates of antimicrobial exposure in these settings. However, this is never explicitly addressed in the manuscript nor are there references provided to support this important issue. This should be addressed at least in the Discussion.
- The authors refer to mobile genetic elements throughout the manuscript but never provide a formal definition. In the literature, plasmids are sometime also considered MGEs (eg. <https://journals.asm.org/doi/10.1128/CMR.00088-17>) and it would be helpful if the authors address this specifically as it is central to the study.
- The Discussion would benefit from some further analysis of how the study is situated within the literature, as well as analysis of some of the shortcomings of the study and possible future directions.
- The authors identify ARGs and MGEs in their analysis of publicly available data on the basis of regular expression matching of GenBank annotations. This has two issues: 1) the GenBank annotations are frequently inaccurate; 2) how did the authors ensure that their regular expressions are a comprehensive representation of ARGs and MGEs? Did the authors do any independent assessment against widely used databases such as the Comprehensive Antibiotic resistance (CARD) database and try to annotate the genomes themselves?

Minor comments:

- Figure 1: While covered in the text, it would be helpful to provide the legends with relevant colors in the figure as they differ between Figure 1A/B and Figure 1C/D. Also would be helpful to have the legend for colors used in Figure 1E.
- Line 116: it would be useful to provide a supplementary data file that shows the included genomes for this analysis and the relevant findings regarding ARGs, as well as some supplementary tables/figures that indicate the basic descriptive details such as the species, ARGs etc. across the different host source categories.
- Line 118: how did the authors ensure that the genomes were truly complete? Was there any assessment of circularization of the chromosomes or plasmids?
- Line 169: Downsampling to one genome may be overly reductive, especially given the diversity amongst different bacterial species. A more data-driven approach such as using Assembly Dereplicator (<https://github.com/rrwick/Assembly-Dereplicator>) may be worthwhile.
- Line 191: Can the authors specify why these particular datasets were chosen? They are relatively small and could be supplemented by a greater range of publicly available datasets (see Major Comments above), which may add weight to the authors' findings.
- Line 205: This may be an overstatement, it may be more appropriate to state that the clinical antibiotic-resistant isolates from the analyzed datasets are enriched with duplicated ARGs. The analyzed datasets are relatively limited and may not be generalizable, as outlined above.
- Line 442: It may be worthwhile to place the description of the clinical antibiotic-resistant datasets after the publicly available datasets (stating line 460) given that this is how the Results section is ordered.
- Line 460: Was there additional quality control done on the downloaded complete genomes to

ensure that they were bacterial chromosomes. Similarly, was there any additional work done to confirm that the downloaded plasmids were actually plasmids?

Responses to reviewers' comments

Reviewer #1 (Remarks to the Author):

The topic is of interest and is timely as AMR is important, and gene duplication is an interesting and poorly studied phenomena because of technical issues. Using new technology of long sequencing reads is a potential solution to support studies of AMR duplication importance.

The abstract could be enhanced by including a statement that the study is multidisciplinary and uses long sequencing reads, experimental data and mathematical modelling.

We thank the reviewer for recognizing the significance of our work, as well as its multidisciplinary nature. We have added the following sentence to the Abstract as suggested by the Reviewer:

Here, we use a combination of modeling and experimental evolution to examine this hypothesis and use long-read genome sequences of tens of thousands of bacterial isolates to examine its generality in nature.

Overall, I found the manuscript interesting. However, I struggled to interpret the figures as they were missing key information. Importantly, the rationale of study design was often not clear. And this meant that most of the main conclusions were not supported in the current format.

We appreciate the constructive feedback and have addressed each specific comment made by the Reviewer below and have revised the manuscript accordingly.

Specifically -

1. Mathematical modelling can be useful here, although I couldn't seem to find the assumptions that were used in Figure 1B. The results are highly dependent on the assumptions made. What is the strong antibiotic selection and high cost of expression that is plotted here? Line 89-90 does not seem to be proven by Figure 1B?

We have added the following text to the Material and Methods section on the mathematical model,

Unless otherwise stated, the simulation results shown in Figure 1 use the following default parameter settings (arbitrary units): Antibiotic concentration $A = 2.0$, Duplication cost $c = 0.1$, Transposition rate $\eta = 0.0002$, Dilution rate $D = 0.1$, Plasmid copy number $y = 2$.

We have added the following text to the Figure caption for Figure 1B:

The simulation result in this panel uses the following parameter settings (arbitrary units): Antibiotic concentration $A = 2.0$, Duplication cost $c = 0.1$, Transposition rate $\eta = 0.0002$, Dilution rate $D = 0.1$, Plasmid copy number $y = 2$ (Methods: Mathematical model). Under these conditions, the fitnesses of the three subpopulations are ordered $f_1 < f_2 < f_3$.

Lines 103-115 (lines 89-93 in the initial submission) describe how fitness is parameterized in the model. Figure 1B illustrates the evolutionary dynamics for the case that higher ARG expression monotonically confers higher fitness in the presence of antibiotic. We have added a reference to the relevant section of the Methods as follows:

The fitness of each population therefore depends on antibiotic concentration, the cost of ARG expression, and the effective number of ARG copies per cell in each subpopulation, which depends on plasmid copy number (Methods: Mathematical model: Fitness functions).

We noticed and fixed a typo in the Figure 1B legend. It should read:

Under sufficiently strong antibiotic selection and with low cost of expression, cells with ARGs on the plasmid dominate the population.

To clarify the logic of our reasoning, we also added the following sentences (lines 103-115):

Under antibiotic selection, one of the subpopulations with the additional ARG copy rapidly outcompetes the others, depending on which has the highest fitness. When the cost of expressing additional ARG copies is low, the Type 3 subpopulation, which contains duplicated ARGs on the plasmid, dominates (Figure 1B). When the cost of expressing the ARG on the plasmid outweighs the benefit of resistance, the Type 2 subpopulation, which contains duplicated ARGs on the chromosome, dominates (Supplementary File 1). By defining a “Duplication Index” as the fraction of the population with a duplicated ARG, we find that duplicated ARGs rapidly establish throughout the population at a threshold antibiotic concentration. As the cost of ARG expression increases, this threshold concentration increases. This is shown by the rightward shift of curves representing higher ARG expression costs in Figure 1C. In addition, as the transposition rate of the transposable ARG increases, the time for establishment of duplicated ARGs in the population decreases, as shown by a leftward shift of curves representing higher transposition rates in Figure 1D.

2. Figures 1C and 1D, Line 90-93 is dependent on the cost of the ARG expression. What is the evidence for this, or assumptions made when plotting the graphs?

To clarify this point, we have added the following text to the Figure 1C legend:

The simulation result in this panel uses the following parameter settings (arbitrary units): $A = 2.0$, $\eta = 0.0002$, $D = 0.1$, $\gamma = 2$. Colors shift from yellow to blue as the fitness cost of carrying duplicated ARGs increases. The yellow curve represents Duplication cost $c = 0.05$, and each successively darker curve represents an increment of 0.05, up to the darkest curve of $c = 0.25$. See Supplementary File 1 for further details.

We also changed the Figure 1D color scheme to be consistent with Figure 1C (i.e., darker shades = higher parameter values), and edited the Figure 1D legend as follows:

The simulation result in this panel uses the following parameter settings (arbitrary units): $A = 2.0$, $c = 0.1$, $D = 0.1$, $\gamma = 2$. Colors shift from yellow to blue as the transposition rate η increases.

In addition, we have added the following sentences to the main text to clarify the interpretation of Figures 1C and 1D:

As the cost of ARG expression increases, this threshold concentration increases. This is shown by the rightward shift of curves representing higher ARG expression costs in Figure 1C. In addition, as the transposition rate of the transposable ARG increases, the time for establishment of duplicated ARGs in the population decreases, as shown by a leftward shift of curves representing higher transposition rates in Figure 1D.

3. Line 94. What is the evidence that antibiotic selection CAUSES high frequency duplication? Presumably it is just selected for?

We have rewritten this phrase for clarity.

We tested the core prediction of this model— that duplicated ARGs evolve in response to antibiotic selection.

4. Figure 1E. This figure is very difficult to interpret. What do all the listings on the left mean? Are they mutations? Duplications? What does the x-axis mean? How many replicates? The method says 30 times, but only four presented? How was the data captured or interpreted as it represents a population rather than individual cells? Figure 2C is also similarly difficult to interpret.

We appreciate the feedback. We have added labels to the y-axis to indicate that these are genes hit by *de novo* variants (point mutations, indels, insertion element transpositions), and we have added brackets to separate the variants into two classes: transpositions of the tetA-Tn5 transposon, versus all other *de novo* mutations observed. We have also added labels to the x-axis to indicate that these represent populations.

Each column represents an independent replicate population, which we have further clarified in the Methods section:

The next day, 5× replicate populations were inoculated using 3μL of the overnight culture for each ancestral clone. Evolving populations were transferred nine times, increasing tetracycline concentrations in LB on each transfer (2, 4, 6, 8, 10, 20, 30, 40, 50 μg/mL). In total, 30 populations evolved in LB + tetracycline from the six ancestral strains.

To clarify the interpretation of these data, we walk through the results shown in Figures 1E and 2C more explicitly in the main text, to aid readers who are less familiar with the genomic analysis of microbial evolution experiments. See lines 116-147 in the revised manuscript.

5. Figure 2A. Is active meaning the transposon is active? How many replicates were performed, its not possible to tell from the figure. Line 108, what is the evidence for “high frequencies”?

Yes, this means that the transposon is active due to presence of the Tn5 transposase. 24 replicates were performed: 12 with no selection, 12 with tetracycline selection.

To clarify we have added the following text to the Figure 2 legend:

12 replicate populations were evolved for one day under tetracycline selection, and another 12 replicate populations were evolved in LB without antibiotic as a control. Each panel shows a result generated by whole-population Illumina sequencing of these evolved populations.

We have clarified that “high frequencies” means observable frequencies by whole-population Illumina sequencing:

Given the high activity of the synthetic tetA-Tn5 transposon, one day of tetracycline selection (~10 bacterial generations) was sufficient to drive duplications of the tetracycline resistance gene to observable allele frequencies across all replicate populations.

6. Figure 2B. Which method was used to generate the data? What does the y-axis mean? How many replicates?

Please also refer to our response to point 5. We have clarified that one dataset (whole-population Illumina sequencing of evolved populations) underlies all panels of Figure 2, with 12 replicate populations evolved under tetracycline resistance, and 12 replicate populations evolved in LB without antibiotic. The y-axis means the number of variants ("Count") counted in that treatment.

7. Figure 2C. See point 4.

See our response to point 4.

8. Where is the *acr* locus located? What do the colours mean? How many replicates? Line 110-112, what is the evidence this region is *acr* and was it confirmed by any other method/evidence?

The *acr* locus is found between 481254-486408 in the K12 MG1655 NC_000913 reference genome that we used for genome resequencing in this analysis.

Based on this feedback, we have edited the Figure 2D legend as follows:

D. After 1 day of growth under a treatment of 5 ug/mL, all six of the populations that lack active transposase (shown in blue) show chromosomal amplifications around the location of native antibiotic resistance efflux pump operon *acrAB* in the K12 MG1655 NC_000913 reference genome. Populations with Tn5 transposase, or that were not treated with antibiotic, lack these amplifications.

The read mapping in Figure 2D shows an increase in sequencing coverage of the *acr* locus in the reference genome, across the 6 replicate populations that were treated with tetracycline but did not have Tn5 transposase to allow for *tetA* duplications. This analysis depends on the *acr* sequence, and not the specific coordinates of the *acr* locus in the genome. In other words, the copy number inference made by this Illumina read mapping method is robust to large-scale genomic rearrangements that would move *acr* to a different location.

We have also clarified that the blue color is a visual guide for saliency in the figure legend, and we have added the number of replicates in the main text: 3 per treatment, for 12 populations with Tet 5 selection, and 12 populations with no Tet selection.

9. Line 105-114, I cannot see the convincing evidence for these statements in Figure 2. "...drive the evolution of..." seems an overstatement when a single tube with antibiotic resistant bacteria is exposed to antibiotic, its not clear that duplication of genes is required.

We did not observe duplications when cultures were passaged in the absence of antibiotic treatment (Supplementary File 2). By contrast, gene duplications occurred only when the cultures were passaged in the presence of increasing strong antibiotic selection. Thus, our experiments indeed demonstrated antibiotic treatment caused gene duplication. The reviewer is correct that our data do not demonstrate duplication of genes is always required for bacteria to survive the increasing strong antibiotic selection. We did not intend to make this claim.

To better clarify this point, we have added Supplementary File 2, which contains a list of all mutations observed across all our evolution experiments, including the no-antibiotic treatments. In addition, we have edited the main text to clarify the logic and purpose of our experiments. See lines 116-147 in the revised manuscript.

More broadly, our bioinformatic results show that duplicated ARGs are enriched, but not always found, in bacteria isolated from humans and livestock, and clinical antibiotic-resistant isolates. These experiments demonstrate that ARG duplication is one mechanism for the evolution of resistance. Our bioinformatic analysis shows that this mechanism has relevance for clinical and natural microbial populations.

10. Line 117. The bioinformatics study is potentially interesting. Why were these datasets chosen? Where did they come from? Are they long reads? It would be helpful to include estimates of the number of samples compared in each category in the main text/figures.

We have clarified that we downloaded *all* complete and fully annotated bacterial genomes from NCBI RefSeq passing additional quality control checks, and analyzed the subset of these data with ecological metadata.

11. Figure 4A. “specifically associated” is not an accurate description of the data.

We have edited this to: “D-ARGs are specifically enriched in bacterial isolates from humans and livestock”.

12. Line 157-175 is dependent on Figure S3. I think the data shows that known human pathogens that contain more D-ARGs are found in humans and animals, but its difficult to justify the extensive discussion here is robust. Downsampling is figure S3D not C.

Thanks for pointing out this typo. The point of this section is to check the robustness of the main result in Figure 4. This analysis shows that:

- 1) the enrichment pattern is primarily driven by a handful of genera,
- 2) the enrichment of D-ARGs in humans and livestock holds when we only analyze these genera,
- 3) the same pattern holds when we dramatically downsample the data, so that the data are not biased by over-representation of highly similar strains (i.e. with the same duplicated ARGs) in the human and livestock categories.

We added the following topic sentence to emphasize that this section is describing additional computational controls:

We checked the robustness of the pattern shown in Figure 4A with further computational controls. We reasoned that the association between duplicated ARGs and isolates from humans and livestock could be affected by both the over-representation of some bacterial taxa, as well as phylogenetic correlations between highly related isolates.

13. Line 192. Why were these isolates chosen? Are they long reads? Without knowing why they were chosen, the assumptions and conclusions made in this paragraph are invalid.

We chose these isolates for 3 reasons. 1) they are high quality genomes sequenced by long-read technologies 2) we know their provenance as antibiotic-resistant clinical isolates, and 3) they are completely independent from our main dataset from RefSeq, thus allowing for statistical validation.

We have edited this paragraph, to explain our reasoning as follows:

To validate the medical relevance of these findings, we searched the recent literature for datasets of bacterial genomes satisfying three criteria: 1) High-quality, publicly available, complete and fully annotated genomes sequenced by long-read technologies; 2) known provenance from antibiotic-resistant clinical isolates; and 3) independence from our main dataset of complete genomes from RefSeq, to rigorously test the hypothesis that antibiotic treatment selects for duplicated ARGs. We found three genomic datasets satisfying these criteria and measured the extent to which each dataset contained duplicated ARGs.

14. Line 223-226, some statistics are needed for this statement in Figure S8.

We have added these statistics in detail:

Furthermore, duplicated genes encoded solely on plasmids are more likely to encode antibiotic resistance and functions other than those associated with mobile genetic elements (40,714 duplicated genes encoding ARGs and other non-MGE functions, compared to 32,008 duplicated genes encoding MGE functions), in comparison to both duplicated genes encoded solely on the chromosome (297,239 duplicated genes encoding ARGs and other non-MGE functions, compared to 451,951 duplicated genes encoding MGE functions), and duplicated genes encoded on plasmids and the chromosome (31,555 duplicated genes encoding ARGs and other non-MGE functions, compared to 124,615 duplicated genes encoding MGE functions), as shown in Supplementary Figure S11. Therefore, duplicated genes found solely on plasmids have higher proportions of ARGs and other functional genes, in comparison to duplicated genes found solely on chromosomes (Binomial test: $p < 10^{-200}$) and duplicated genes found on both chromosomes and plasmids (Binomial test: $p < 10^{-300}$).

15. Line 238-244 and Figure 5A, 5B. This does not seem to be intuitive. Are there numbers to support these conclusions instead of just proportions? Why would there be so many duplicate MGE genes found on the chromosome?

We did not expect this result before the analysis. However, it does make sense in retrospect. MGE genes include genes like transposases, phage, and integrases, that encode “DNA copy-paste” and “DNA cut-paste” machinery. Thus, it makes sense that genes that facilitate “DNA copy-paste” operations are over-represented among duplicated genes. We count Insertion Sequence transposases (IS elements) as MGE genes, and it is well-known that IS element transpositions/duplications contribute to bacterial adaptation on laboratory timescales. One of us previously found that IS150 duplications are important factors in adaptive evolution in an evolution experiment based on some of Lenski’s LTEE strains (see Section: “Contribution of transposable insertion elements to parallel evolution”, in Blount, Maddamsetti, and Grant et al. in eLife: “Genomic and phenotypic evolution of *E. coli* in a novel citrate-only resource environment”).

As suggested, we have included more precise numbers to supplement the proportions, and we now provide further interpretation of this result in lines 300-311 of the revised manuscript.

16. Line 290. The opening sentence is not really justified by the data. Line 297 is not proven either. Duplication is not the same as location next to an MGE gene.

We have edited the opening sentence of the Discussion to clarify our logic and its connection to our data:

Our modeling and experiments demonstrate that ARG duplication is an effective mechanism for the evolution of antibiotic resistance. Our genomic analyses show that MGEs, such as the transposons and plasmids in our experiments, can serve as a vehicle for the duplication of ARGs.

We elaborate further on the reasoning for the claim on line 297 (of our initial submission), but the main justification for the claim comes from the evolutionary genomics literature on gene duplication (we cite a couple key references here):

Together, these results imply that antibiotic usage not only enriches for resistant subpopulations, but also selects for mutants with a higher capacity for evolutionary innovation through gene duplication. With duplication, one gene copy can maintain ancestral function, while additional copies can evolve new functions (1, 6).

We also clarify that we mean ARGs carried *within* MGEs. Such ARGs are duplicated when the entire MGE copies itself somewhere else.

Our results indicate that MGEs have an intrinsic ability to drive evolutionary innovation through their ability to catalyze the duplication and HGT of passenger genes, such as ARGs, carried within the MGE (52, 53).

17. Could the authors differentiate between duplication of an MGE and duplication of an ARG? How often do they both occur together? Or singly? This may help interpret their conclusions? Is it suggested that MGE duplication rather than ARG duplication is the major mechanism of duplication?

MGE duplication is one but not the only mechanism for ARG duplication. Recombination between short DNA repeats can also cause duplications of antibiotic resistance genes (20). The relative contributions of the various mechanisms of the gene duplication (i.e. amplification of tandem repeats versus MGE transposition) in the evolution of antibiotic resistance is not well understood.

We agree that it would be interesting and important to measure the extent to which MGE duplications/transpositions contribute to ARG duplications and spread. We believe that this is an important research gap for the field, given its technical challenges, and we now say so in the Discussion in lines 376-381 of our revised manuscript.

Reviewer #2 (Remarks to the Author):

In this study, Maddamsetti and colleagues investigate the relationship between horizontal gene transfer and gene duplication as a driver of the spread of antimicrobial resistance (AMR). This well-conducted study is novel and has multiple sequential components including mathematical modelling, in vitro lab experiments and analysis of a large genomic dataset that together provide an interesting hypothesis for a previously overlooked aspect of AMR spread.

We thank the reviewer for recognizing the novelty and quality of our study.

The major issues with the study are:

- The authors conduct in vitro experiments in one species (*E. coli*) and one resistance determinant (*tetA*) and use this to confirm their mathematical modelling findings. I am not sure that this is sufficient evidence to support their claims. It may be worthwhile considering further experiments in other relevant human pathogens with significant burdens of AMR (eg. *Klebsiella pneumoniae*) and more clinically relevant resistance determinants (eg. beta-lactamases).

We appreciate the suggestion. Due to biosafety concerns, we are not ready to introduce additional antibiotic resistance to known pathogens and to evolve stronger resistance. In light of the reviewer's suggestion, however, we conducted additional experiments with our model system, which show that our findings indeed generalize to resistance determinants beyond *tetA*, including beta-lactamases (see the new Supplementary Figure 1). Thus, our experimental results, along with the extensive bioinformatic analysis, demonstrate that antibiotic-mediated ARG duplication (through MGEs or other mechanisms) can readily happen in nature and in laboratory evolutionary experiments.

- The authors recognize that an inability to detect plasmid copy number variation is a limitation of their analyses of publicly available genome data. However, the authors have access to three clinical datasets which have both genome assemblies and read datasets available, as well as potentially numerous other public datasets) (eg. <https://www.ncbi.nlm.nih.gov/pmc/articles/PMC9396894/>). This should make it possible to provide some estimates of the role of plasmid copy number variation.

We appreciate the suggestion. We conducted further analysis of plasmid copy number in a set of 114 complete sequences of genomes of antimicrobial resistant organisms from an Australian ICU (NCBI BioProject PRJNA646837). This analysis shows that plasmid copy number can indeed increase ARG copy number (Supplementary Figure 10).

The original three clinical datasets in our dataset were sequenced with both long reads (Oxford Nanopore) and Illumina short read technology, but were not deposited in NCBI RefSeq as complete genomes with resolved plasmids. For this reason, we did not analyze plasmid copy number in those data.

- The authors imply throughout the study that isolates from human and livestock carry an increased number of duplicated ARGs due to the higher rates of antimicrobial exposure in these settings. However, this is never explicitly addressed in the manuscript nor are there references provided to support this important issue. This should be addressed at least in the Discussion.

We have added the following text to the Discussion:

The enrichment we observe of duplicated ARGs in humans and livestock is most likely caused by high rates of antimicrobial exposure (58). Indeed, our analysis of clinical antibiotic-resistant strains strongly supports antibiotic use as a primary driver for the evolution of duplicated ARGs, even though we do not know the resistance phenotypes or antibiotic treatment history for most of the genomes in this study. Future research could examine the quantitative relationship between antibiotic use and the evolution of duplicated ARGs in settings such as hospitals (59) and factory farms (60, 61).

- The authors refer to mobile genetic elements throughout the manuscript but never provide a formal definition. In the literature, plasmids are sometime also considered MGEs (eg. <https://journals.asm.org/doi/10.1128/CMR.00088-17>) and it would be helpful if the authors address this specifically as it is central to the study.

We have added the following text to the Introduction to give a more concrete definition, following the reference provided by the Reviewer.

Following Partridge et al. (21), we define MGEs as “elements that promote intracellular DNA mobility (e.g., from the chromosome to a plasmid or between plasmids) as well as those that enable intercellular DNA mobility”. In our experiments, we focus on transposons and plasmids, which are known to mediate the horizontal transfer of ARGs in microbial communities (5, 22). Our bioinformatics analyses more broadly examine genes encoding MGE components, including genes involved in transposon, integrase, bacteriophage, and plasmid functions.

- The Discussion would benefit from some further analysis of how the study is situated within the literature, as well as analysis of some of the shortcomings of the study and possible future directions.

We appreciate the suggestion. We have revised the Discussion to discuss the limitations of this study and potential future directions:

- 1) The need for databases/accurate annotation of MGEs across bacterial genomes
- 2) The need to quantify antibiotic use and the evolution of duplicated ARGs in natural and clinical isolates.

- The authors identify ARGs and MGEs in their analysis of publicly available data on the basis of regular expression matching of GenBank annotations. This has two issues: 1) the GenBank annotations are frequently inaccurate; 2) how did the authors ensure that their regular expressions are a comprehensive representation of ARGs and MGEs? Did the authors do any independent assessment against widely used databases such as the Comprehensive Antibiotic resistance (CARD) database and try to annotate the genomes themselves?

Thanks for raising these important points.

To ensure high-quality annotations, we have revised our analysis to focus on genomes deposited in RefSeq. Li et al. (40) report in *Nucleic Acids Research* that changes in the Prokaryotic Genome Annotation Pipeline (PGAP) since 2018 have resulted in a substantial reduction in spurious annotation in RefSeq.

As an additional control, we also annotated ARGs and MGE-associated genes using homology to the CARD database and the mobileOG-db database of protein families associated with mobile genetic elements. We find that our results hold when using homology to these databases, instead of regular expressions, to classify ARGs and MGE-associated genes (Supplementary Figures 14, 15, 16). Following the computational protocol in Gibson et al. (74), we classified homologs in our RefSeq dataset based on >80% identity over >85% of the target sequences in CARD and mobileOB-db.

In addition, we used the results of these homology searches to assess the precision and recall of our regular expressions to capture ARGs and MGEs. Our ARG regular expression has a precision of 93% and a recall of 97%, based on the duplicated ARGs found by homology to CARD, and our MGE regular

expression has a precision of 79% and a recall of 87%, based on the duplicated MGE-associated genes found by homology to mobileOG-db.

Minor comments:

- Figure 1: While covered in the text, it would be helpful to provide the legends with relevant colors in the figure as they differ between Figure 1A/B and Figure 1C/D. Also would be helpful to have the legend for colors used in Figure 1E.

Thanks for the feedback. We have made these improvements to Figure 1.

- Line 116: it would be useful to provide a supplementary data file that shows the included genomes for this analysis and the relevant findings regarding ARGs, as well as some supplementary tables/figures that indicate the basic descriptive details such as the species, ARGs etc. across the different host source categories.

We have added Supplementary File 3, which lists the genomes in the main bioinformatics analysis, their ecological annotation, and whether they contain duplicated ARGs. We have also added Supplementary File 4, which lists all duplicated ARGs, and the genomes in which they are found.

- Line 118: how did the authors ensure that the genomes were truly complete? Was there any assessment of circularization of the chromosomes or plasmids?

Thanks for raising this important point. In our original submission, we included both complete genomes and some genomes containing assembled chromosomes and plasmids, allowing for some unplaced scaffolds. We have done additional quality control, to ensure both completeness and correctness, by ensuring that all genomes passed Genbank QC for deposition into RefSeq, and by ensuring that all genomes have no gaps within scaffolds or between scaffolds, i.e. circularization. In addition, we have removed all samples with unplaced scaffolds from the analysis. We now report these details in the Methods.

- Line 169: Downsampling to one genome may be overly reductive, especially given the diversity amongst different bacterial species. A more data-driven approach such as using Assembly Dereplicator (<https://github.com/rwick/Assembly-Dereplicator>) may be worthwhile.

Thanks for this suggestion. We have included an additional analysis using Assembly Dereplicator (Supplementary Figure 4D).

- Line 191: Can the authors specify why these particular datasets were chosen? They are relatively small and could be supplemented by a greater range of publicly available datasets (see Major Comments above), which may add weight to the authors' findings.

We have added more details for why we chose these datasets: see our response to point 13 raised by Reviewer 1.

Thanks for pointing us to the additional dataset of WGS data of antimicrobial organisms from an Australian ICU (<https://www.ncbi.nlm.nih.gov/bioproject/PRJNA646837>). We have analyzed these data, and incorporated them into our revised manuscript (Supplementary Figure 9).

- Line 205: This may be an overstatement, it may be more appropriate to state that the clinical antibiotic-

resistant isolates from the analyzed datasets are enriched with duplicated ARGs. The analyzed datasets are relatively limited and may not be generalizable, as outlined above.

We have qualified our statement as suggested:

Therefore, the clinical antibiotic-resistant isolates in these additional datasets are enriched with duplicated ARGs, relative to the general human isolates in our main dataset (Binomial test: $p < 10^{-8}$).

- Line 442: It may be worthwhile to place the description of the clinical antibiotic-resistant datasets after the publicly available datasets (stating line 460) given that this is how the Results section is ordered.

We have made this suggested change.

- Line 460: Was there additional quality control done on the downloaded complete genomes to ensure that they were bacterial chromosomes. Similarly, was there any additional work done to confirm that the downloaded plasmids were actually plasmids?

Each downloaded complete genome (now from NCBI RefSeq rather than Genbank) contains at least one chromosome and may contain one or more plasmids in the assembly. Therefore, every downloaded plasmid is associated with at least one chromosome, corresponding to the genome in which it is found, and we ran an additional test to verify this assertion in our analysis. We also checked each plasmid to make sure that they are smaller than their corresponding chromosome, to catch misannotation errors in which plasmids and chromosomes are switched. Finally, we annotated all plasmids in our data with MOB-typer (Supplementary File S5). We now include these details in the Methods.

Reviewers' Comments:

Reviewer #2:

Remarks to the Author:

Thank you for the revision of the manuscript. My queries have been addressed and I have no further suggestions.

Reviewer #3:

Remarks to the Author:

The interdisciplinary work presented by the authors in this manuscript is convincing to raise relevant and interesting hypotheses on gene duplication and AMR. The authors have made substantial efforts to answer previous reviewer comments. I do not have any new major comments on the current format, although I would like to raise a few follow-up points which I think should be addressed to reinforce the manuscript:

- Figure 1: the colours for type 2/3 are inversed between 1A and 1B

- I agree with comment 1) by reviewer 1 on the dependence of the modelling results on the assumptions made. This is by itself not a problem, as long as one is clear about those assumptions, and I still don't think this is the case. Currently, all the modelling results rely on one set of parameter values, but without any justification or source for these values. If these correspond to arbitrary yet "realistic" values, this must be clearly stated and explained. The presentation and discussion of the modelling results must be adjusted to clarify that these results only highlight what the dynamics of gene duplication and antibiotic selection could look like in certain conditions, providing a basic hypothesis to test in vitro instead of providing direct answers. For example, in the first paragraph of the Discussion, it would be more accurate to rephrase to "Our modelling demonstrates how ARG duplication could be an effective mechanism...", instead of the current strong phrasing.

- Ideally however, it would be beneficial to discuss how varying antibiotic concentration and transposition rates could affect these dynamics, with the help of a supplementary figure on the topic. This would then allow you to more confidently discuss the overall system dynamics, and bring further value to the modelling results.

- I think your alternative phrasing for line 94 in response to reviewer 1 comment 3) still implies a link of causality not supported by evidence, and would instead suggest you replace "evolve in response to" by "are selected for by"

- Following reviewer 2 comments 1) and 5), I think you should clearly state as a limitation in your Discussion that you were only able to focus on E. coli. Would you expect your results to be valid for other bacterial species, and why/why not? What differences, if any, would you expect, perhaps based on what is known about the prevalence of MGEs in other species than E. coli? Your Discussion is currently very brief and would benefit from further content.

- Similarly, although I appreciate that you were able to repeat your experiment with different genes than tetA, your Discussion would benefit from a paragraph on the value of looking at these different genes, and the remaining limitations if any (e.g. what type of resistance mechanism is encoded by each gene, efflux pump, target modification, antibiotic degradation etc, and how could this affect your results)

Responses to reviewers' comments

Reviewer #2 (Remarks to the Author):

Thank you for the revision of the manuscript. My queries have been addressed and I have no further suggestions.

We thank the reviewer for their valuable critical feedback, which substantially improved our paper.

Reviewer #3 (Remarks to the Author):

The interdisciplinary work presented by the authors in this manuscript is convincing to raise relevant and interesting hypotheses on gene duplication and AMR. The authors have made substantial efforts to answer previous reviewer comments. I do not have any new major comments on the current format, although I would like to raise a few follow-up points which I think should be addressed to reinforce the manuscript:

We thank the reviewer for recognizing the significance of our work, as well as its interdisciplinary nature. We have revised our manuscript in response to the follow-up points that they raised.

- Figure 1: the colours for type 2/3 are inversed between 1A and 1B

Thank you for pointing this out, we have fixed this error (colors were correct but "Type 2" and "Type 3" labels were incorrectly swapped in panel 1B).

- I agree with comment 1) by reviewer 1 on the dependence of the modelling results on the assumptions made. This is by itself not a problem, as long as one is clear about those assumptions, and I still don't think this is the case. Currently, all the modelling results rely on one set of parameter values, but without any justification or source for these values. If these correspond to arbitrary yet "realistic" values, this must be clearly stated and explained. The presentation and discussion of the modelling results must be adjusted to clarify that these results only highlight what the dynamics of gene duplication and antibiotic selection could look like in certain conditions, providing a basic hypothesis to test in vitro instead of providing direct answers. For example, in the first paragraph of the Discussion, it would be more accurate to rephrase to "Our modelling demonstrates how ARG duplication could be an effective mechanism...", instead of the current strong phrasing.

- Ideally however, it would be beneficial to discuss how varying antibiotic concentration and transposition rates could affect these dynamics, with the help of a supplementary figure on the topic. This would then allow you to more confidently discuss the overall system dynamics, and bring further value to the modelling results.

Thanks for this suggestion. We have added this analysis, which is now in the new Figure 1E, and we now qualify the modeling results, by adding the following text (lines 115-119):

Furthermore, the model shows that for any given ARG expression cost, duplicated ARGs will establish in the population when both the transposition rate and antibiotic concentration are sufficiently high (Figure 1E). Altogether, these results highlight what the dynamics of antibiotic selection and ARG duplication could look like, and illustrate a basic model that can be tested experimentally.

And as suggested, we have rephrased the first sentence of the Discussion to read:

“Our modeling demonstrates that ARG duplication could be an effective mechanism for the evolution of antibiotic resistance”.

- I think your alternative phrasing for line 94 in response to reviewer 1 comment 3) still implies a link of causality not supported by evidence, and would instead suggest you replace "evolve in response to" by "are selected for by"

We have edited this phrase as follows:

We tested the core prediction of this model— that antibiotics select for duplicated ARGs—

In addition, we edited this subsection heading from “Duplicated ARGs evolve in response to antibiotic selection” to “Antibiotics select for duplicated ARGs”.

- Following reviewer 2 comments 1) and 5), I think you should clearly state as a limitation in your Discussion that you were only able to focus on *E. coli*. Would you expect your results to be valid for other bacterial species, and why/why not? What differences, if any, would you expect, perhaps based on what is known about the prevalence of MGEs in other species than *E. coli*? Your Discussion is currently very brief and would benefit from further content.

- Similarly, although I appreciate that you were able to repeat your experiment with different genes than *tetA*, your Discussion would benefit from a paragraph on the value of looking at these different genes, and the remaining limitations if any (e.g. what type of resistance mechanism is encoded by each gene, efflux pump, target modification, antibiotic degradation etc, and how could this affect your results)

We appreciate the suggestion. We address both points in the following paragraph that we added to the Discussion:

Our analysis has several caveats that warrant analysis in future research. First, our mathematical model implicitly assumes that the mutation rate for genomic resistance mutations is small compared to ARG transposition rates— small enough that genomic resistance mutations can be ignored. More work is needed to measure how the relative magnitudes of these rates, and the relative selective benefits of these molecular mechanisms, affects the evolution of antibiotic resistance by ARG duplication. Second, our experiments focused on *E. coli*, and did not examine whether MGEs promote gene duplication across bacterial species. Given our bioinformatics results, we expect our experimental findings to hold across bacteria, but direct experimental tests are needed. For instance, we expect that the transposition rates may depend on idiosyncratic interactions between a given MGE and its host. In this case, it would be interesting to ask whether the prevalence of a given MGE in a particular bacterial species can predict the transposition rate of that MGE in that species, and thus the importance of particular MGEs for spreading clinically relevant resistance in different pathogen species. Third, while we examined several different antibiotic resistance genes in our experiments, it is unclear whether the type of resistance mechanism encoded by an ARG (e.g. antibiotic degradation, target modification, efflux pump activity) affects the likelihood of resistance evolution by gene duplication. Given the generality of our bioinformatic results across multiple classes of antibiotics (Supplementary Figure 2), we predict that the specific molecular mechanism of resistance has little impact on ARG duplication dynamics. Indeed, our mathematical model predicts that the dynamics of duplicated ARGs only depends on transposition rates and the balance of fitness benefits and costs of expressing duplicated ARGs, which needs to be tested by future experiments with a broader set of ARGs operating with different mechanisms.